# ConStat: Performance-Based Contamination Detection in Large Language Models

**Jasper Dekoninck**[1]**, Mark Niklas Müller**[1,2]**, Martin Vechev**[1]

Department of Computer Science[1]
ETH Zurich, Switzerland
{jasper.dekoninck,martin.vechev}@inf.ethz.ch

LogicStar.ai[2]

mark@logicstar.ai

## Abstract

Public benchmarks play an essential role in the evaluation of large language models. However, data contamination can lead to inflated performance, rendering them unreliable for model comparison. It is therefore crucial to detect contamination and estimate its impact on measured performance. Unfortunately, existing detection methods can be easily evaded and fail to quantify contamination. To overcome these limitations, we propose a novel definition of *contamination as artificially inflated and non-generalizing benchmark performance* instead of the inclusion of benchmark samples in the training data. This perspective enables us to detect *any* model with inflated performance, i.e., performance that does not generalize to rephrased samples, synthetic samples from the same distribution, or different benchmarks for the same task. Based on this insight, we develop CONSTAT, a statistical method that reliably detects and quantifies contamination by comparing performance between a primary and reference benchmark relative to a set of reference models. We demonstrate the effectiveness of CONSTAT in an extensive evaluation of diverse model architectures, benchmarks, and contamination scenarios and find high levels of contamination in multiple popular models including MISTRAL, LLAMA, YI, and the top-3 Open LLM Leaderboard models.[1]

## 1 Introduction

As large language models (LLMs) become increasingly effective at a wide range of tasks, many companies and research institutions compete to develop better models [2, 5, 28, 36]. To facilitate this development, a variety of benchmarks have been proposed that allow a standardized in-depth comparison of model performance across diverse tasks [15, 16, 26, 33].

**Data Contamination** Modern LLMs are trained on vast amounts of internet-sourced data, raising the risk of unintentionally including benchmark samples in the training set. Such *data contamination* can lead to artificially inflated benchmark performance that does not accurately reflect a model's true ability to generalize to unseen tasks. However, model providers argue that the impact of this contamination on model performance is negligible [2, 14, 36] and the enormous size of current training sets almost guarantees contamination to some extent. This casts doubt on the relevance of this traditional definition of contamination in the context of LLMs.

**This Work: A New Perspective on Data Contamination** We propose a new perspective on contamination, defining it based on its effect on model performance rather than its cause. Specifically, we *define contamination as artificially inflated, non-generalizing performance*, i.e., we say a model is contaminated if and only if its performance relative to other models is significantly higher on the original benchmark than on a similar reference benchmark. This definition captures the essence of the contamination problem, i.e., performance measurements becoming unreliable for model comparisons.

---

[1]Code available at https://github.com/eth-sri/ConStat.

38th Conference on Neural Information Processing Systems (NeurIPS 2024).

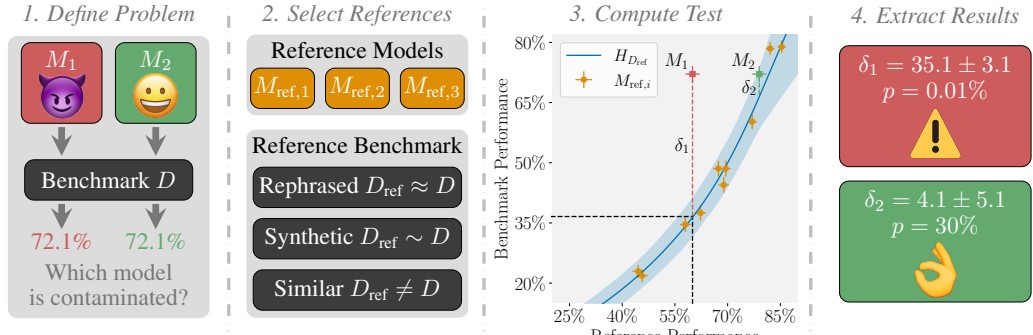

Figure 1: Overview of our method. We first select models to check for contamination, then select reference models and benchmarks, and finally compute CONSTAT to detect and quantify contamination.

Furthermore, it enables principled detection methods that are robust against evasion attacks by malicious providers as this would require generalizing performance improvements.

**Traditional Contamination Detection**  Existing contamination detection methods [18, 23, 24, 31, 34, 37, 40, 41, 49] aim to detect the inclusion of benchmark samples in the training data as a measure of contamination. However, these approaches show limited success, cannot quantify the contamination's effect on model performance, and have to make strict assumptions about the contamination process, making them easy to evade [17].

**This Work: A Statistical Test for Contamination**  In contrast, we leverage our novel performance-based definition of data contamination to propose a statistical contamination test called CONSTAT, illustrated in Fig. 1. Given a target model ($M_1$ or $M_2$) to check for contamination (first step in Fig. 1), we select a set of reference models for performance comparison and a reference benchmark $D_{\text{ref}}$ that is similar to the original benchmark $D$ (second step). This reference benchmark can be a rephrased version of the original benchmark, a synthetic benchmark generated from the same distribution, or a different benchmark measuring performance on the same task. We then evaluate the reference models on both benchmarks $D$ and $D_{\text{ref}}$ and fit the difficulty correction function $H_{D_{\text{ref}}}$ describing the relation between performance on the reference and original benchmarks (blue curve). By evaluating $H_{D_{\text{ref}}}$ at the target model's performance on the reference benchmark, we predict its expected performance on the original benchmark (third step). Finally, we compute the difference $\delta$ between this expected performance and the model's actual performance on the original benchmark. Using bootstrapping, we obtain an estimate of the contamination magnitude $\delta$ and a p-value that quantifies the likelihood of the observed performance difference under the null hypothesis that the target model is not contaminated (fourth step). In the illustrated case, model $M_1$ achieves 60% on the reference benchmark, which translates to an expected performance of 37% on the original benchmark. However, the measured performance of 72% indicates a large contamination effect $\delta_1 = 35\%$ and thus strong contamination with a p-value of 0.01%. In contrast, model $M_2$ shows no signs of contamination.

**Evaluation**  We evaluate CONSTAT on a wide range of contamination scenarios and model architectures, demonstrating that it is significantly more effective at detecting contamination than any prior method. We then use CONSTAT to study a range of popular open and proprietary models and find high levels of contamination in MISTRAL-7b-v0.1 [28], LLAMA-3-70b [2], LLAMA-2-INSTRUCT-70b [43], YI-34b [53], and a range of top Open LLM Leaderboard [7] models.

**Key Contributions**  Our key contributions are:

- We propose a new performance-based definition of benchmark contamination (§2).
- We introduce CONSTAT, a statistical test that detects and quantifies contamination in language models (§3).
- We empirically demonstrate CONSTAT's effectiveness in an extensive evaluation across various contamination scenarios (§4.2).
- We leverage CONSTAT to study a range of popular models and find contamination for MISTRAL, LLAMA, YI, and the top-3 Open LLM Leaderboard models (§4.3-§4.5).

## 2 Defining Contamination

Before formalizing our novel definition, we first informally contrast the traditional, information-flow-based perspective on contamination with our novel performance-based one.

**Information-Flow Perspective**   In traditional machine learning, contamination typically refers to any information flow between the benchmark used for performance measurement and model training. In the context of LLMs, this is usually restricted to the direct inclusion of test set samples (or their semantic equivalents) in the training dataset [37, 39, 51, 56].

However, this perspective suffers from several drawbacks. First, it does not fully capture the core issue of contamination, which is whether it renders test set performance an unreliable predictor of real-world performance. Second, in the era of zero-shot learning, we aim to measure performance on "unseen" tasks, yet we train on internet-scale data that likely contains samples of almost any task. This makes the threshold for contamination blurry. Third, limiting the definition to test sample inclusion neglects the possibility of model and hyperparameter selection based on benchmark performance as a source of contamination. Finally, even with this narrow definition, detecting contamination without access to the training data is challenging, which makes it easy to circumvent [17].

**Performance Perspective**   To overcome these limitations, we propose to *define contamination based on its outcome, rather than its cause*. Informally, we define contamination as artificially inflated performance on a benchmark that does not generalize to real-world performance on the corresponding task, regardless of how it was achieved. This definition aligns better with the practical implications of contamination and enables a more principled detection method that makes evasion difficult.

To detect contamination, we compare the performance of a model $M$ on a benchmark $D$ to its performance on a reference benchmark $D_{\text{ref}}$, the choice of which we will discuss later. It is crucial to account for differences in difficulty between $D$ and $D_{\text{ref}}$. Otherwise, a slightly harder reference benchmark $D_{\text{ref}}$ would falsely indicate inflated performance on $D$. Thus, direct performance comparison is only valid if the distribution over sample difficulties is the same for both benchmarks, which is a very strong assumption that is rarely true. To address this, we compare performances relative to a set of reference models, allowing us to determine if a model's performance on $D$ is significantly higher than expected, given its difficulty. In the next section, we make this definition more formal.

### 2.1 Formal Definition of Performance-Based Contamination

**Reference Models**   To accurately compare performance between benchmarks, we use reference models to correct for benchmark difficulty differences. For this purpose, we consider the set of all reliable LLMs $\mathcal{M}_{\text{ref}}$ from reputable sources to estimate the performance distribution of uncontaminated models. Although we cannot guarantee these models are uncontaminated, we can perform leave-one-out contamination detection to remove suspicious models from the reference set. Furthermore, including contaminated models in $\mathcal{M}_{\text{ref}}$ will only make our test more conservative, making it *less* likely for uncontaminated models to be detected as contaminated.

**Contamination Detection**   For each benchmark $D$, we define a scoring function $S_D \colon \mathcal{M} \to \mathbb{R}$ that assigns a score (e.g., accuracy) to every model from the space of all possible language models $\mathcal{M}$. Applied to the reference models $\mathcal{M}_{\text{ref}}$, it induces a cumulative distribution function $F_D$ over the uncontaminated performance on this benchmark.

We now use the cumulative distributions $F_D$ and $F_{D_{\text{ref}}}$ to predict the performance of a model $M$ on $D$ given its performance on $D_{\text{ref}}$. Specifically, we first map the performance on the reference data $S_{D_{\text{ref}}}(M)$ to a percentile $q = F_{D_{\text{ref}}}(S_{D_{\text{ref}}}(M))$ and then map this percentile to the corresponding performance on the original benchmark $F_D^{-1}(q)$ using the percentile function $F_D^{-1}$. To simplify notation, we define the hardness correction function $H_{D_{\text{ref}}} \colon \mathbb{R} \to \mathbb{R}$ as $H_{D_{\text{ref}}} = F_D^{-1} \circ F_{D_{\text{ref}}}$. This allows us to estimate the effect of contamination on the model's performance as $S_D(M) - H_{D_{\text{ref}}}(S_{D_{\text{ref}}}(M))$ and gives our formal definition of contamination:

**Definition 1** ($\delta$-Contamination)**.** *A model $M \in \mathcal{M}$ is $\delta$-contaminated on a benchmark $D$ with respect to a reference benchmark $D_{ref}$ if $S_D(M) - H_{D_{ref}}(S_{D_{ref}}(M)) > \delta$.*

## 2.2 Types of Contamination

Depending on the choice of reference benchmark $D_{\text{ref}}$, we can measure different types of contamination, depending on how poorly the inflated performance generalizes.

*Syntax-Specific Contamination* occurs when the model fails to generalize to semantically equivalent samples. That is, the model has memorized the exact samples in the benchmark, and its performance drops as soon as the wording changes. We therefore consider it to be the worst kind of contamination. To measure syntax-specific contamination we create our reference benchmark $D_{\text{ref}}$ by rephrasing the samples in the original benchmark $D$ to obtain a semantically equivalent benchmark.

*Sample-Specific Contamination* occurs when the model fails to generalize to new samples from the benchmark distribution. That is, while the model generalizes to samples that are semantically equivalent to those in the original benchmark, it does not generalize to new samples from the same distribution. To accurately measure sample-specific contamination, we would preferably generate samples for $D_{\text{ref}}$ following the same steps used to produce $D$. As this is often infeasible in practice, we instead generate synthetic samples for $D_{\text{ref}}$ by querying a strong LLM using few-shot prompting and varying the provided few-shot examples to increase diversity.

*Benchmark-Specific Contamination* occurs when the model fails to generalize to different benchmarks that aim to measure performance on the same task. That is, the model generalizes to new samples from the original benchmark distribution but does not generalize to closely related benchmarks. To measure benchmark-specific contamination we create (or select) a different benchmark $D_{\text{ref}}$ (e.g., MathQA) that aims to measure performance on the same task as $D$ (e.g., GSM8k). We note that benchmark-specific contamination is by far the least severe type of contamination. Further, while strong sensitivity to the exact benchmark is undesirable, it is important to recognize that even small differences between benchmarks can impact model performance. Therefore, benchmark-specific contamination requires a more nuanced interpretation that takes into account these differences.

## 3 CONSTAT: A Statistical Test for Detecting Contamination

We now present CONSTAT, a novel method for detecting contamination as defined in §2 by computing confidence bounds on the estimated contamination effect using a statistical test.

**Reference Models**   To approximate the underlying distribution of reference models $\mathcal{M}_{\text{ref}}$, we select a diverse sample of $m$ models $\tilde{\mathcal{M}}_{\text{ref}} = \{M_{\text{ref},1}, ..., M_{\text{ref},m}\} \subset \mathcal{M}_{\text{ref}}$. We additionally include an inherently uncontaminated random-guessing model to extend the coverage of our reference set.

**Null Hypothesis**   To rigorously test for contamination, we derive a null hypothesis based on our definition of contamination. The null hypothesis is the assumption that the model $M$ is not contaminated, meaning its actual score on the original data is at most $\delta$ worse than the predicted one: $S_D(M) - H_{D_{\text{ref}}}(S_{D_{\text{ref}}}(M)) \leqslant \delta$ where $\delta \in \mathbb{R}_{\geq 0}$ can be chosen freely.

**Estimating the Hardness Correction Function**   To compute the hardness correction function $H_{D_{\text{ref}}}$, we first estimate the CDFs $F_D$ and $F_{D_{\text{ref}}}$ as the empirical CDFs $\tilde{F}_D$ and $\tilde{F}_{D_{\text{ref}}}$, respectively. To this end, let $i_1, ..., i_n$ be an index such that $S_D(M_{\text{ref},i_k}) \leqslant S_D(M_{\text{ref},i_{k+1}})$. We obtain the CDF $\tilde{F}_D$ as

$$\tilde{F}_D(x) = \begin{cases} 0 & \text{if } x < S_D(M_{i_1}) \\ k/n & \text{if } S_D(M_{i_k}) \leqslant x < S_D(M_{i_{k+1}}) \\ 1 & \text{if } S_D(M_{i_n}) \leqslant x \end{cases} . \tag{1}$$

Similarly, $\tilde{F}_{D_{\text{ref}}}$ can be obtained from an index $j_1, ..., j_n$ such that $S_{D_{\text{ref}}}(M_{\text{ref},j_k}) \leqslant S_{D_{\text{ref}}}(M_{\text{ref},j_{k+1}})$. Using Eq. (1), we find that $H_{D_{\text{ref}}}(S_{D_{\text{ref}}}(M_{j_k})) = S_D(M_{i_k})$. Applying the empirical CDFs directly to other points $x \in [0, 1]$ would result in a step function estimate of $H_{D_{\text{ref}}}$, leading to an overly rough approximation of the hardness correction function. Thus, we compute the approximate hardness function $\tilde{H}_{D_{\text{ref}}}$ by fitting the points $(S_{D_{\text{ref}}}(M_{j_k}), S_D(M_{i_k}))$ using a smoothing spline, minimizing the following loss function:

$$\sum_{k=1}^{n} \left(S_D(M_{i_k}) - \hat{H}_{D_{\text{ref}}}(S_{D_{\text{ref}}}(M_{j_k}))\right)^2 + \lambda \int_0^1 \hat{H}''_{D_{\text{ref}}}(x)^2 \, dx \tag{2}$$

where $\lambda$ is a smoothing parameter that is chosen using generalized cross-validation [46].

**Significance Estimation**   We determine the statistical significance for rejecting the null hypothesis via bootstrapping over both the reference models and the samples in the benchmark, using pivotal intervals [42] to correct for uncertainty in the bootstrapping process. By bootstrapping the models, we consider the effect of our reference model selection $\hat{\mathcal{M}}_{\text{ref}}$. By bootstrapping the samples, we additionally include the error in our estimation of the scores themselves. Thus, given the estimate $\hat{\delta} = S_D(M) - \hat{H}_{D_{\text{ref}}}(S_{D_{\text{ref}}}(M))$ and corresponding bootstrap estimates $\hat{\delta}_1, ..., \hat{\delta}_n$, we compute the p-confidence lower bound for $\delta$ as $\hat{\delta}_{1-p} = 2\hat{\delta} - \hat{\delta}'_{1-p}$ where $\hat{\delta}'_q$ is the $q$-quantile of $\hat{\delta}_1, ..., \hat{\delta}_n$. From this, we obtain the p-value by inverting this lower bound with respect to $q$. Thus, we reject the null hypothesis for a given $\delta$ with significance level $p$ by computing the lowest $p$ such that $2\hat{\delta} - \hat{\delta}'_{1-p} \geqslant \delta$.

**Threat Model**   In accordance with [17], we briefly outline the threat model assumed by CONSTAT. Since we only require the ability to measure the performance of the model on the benchmark, our method is a black-box benchmark-level detection method that is robust to semantic preserving operations. Furthermore, we make no additional assumptions on potential metadata contamination. However, we do rely on the existence of reference models which we can use to estimate the performance of uncontaminated models. Notably, however, we do not assume these reference models to have a similar performance or architecture as the model we wish to test.

# 4   Evaluation

In this section, we evaluate CONSTAT empirically. We first demonstrate CONSTAT's effectiveness, showing it outperforms prior methods in detecting and quantifying contamination across a range of intentionally contaminated models (§4.2). Next, we investigate the contamination of our chosen reference models (§4.3), popular model families (§4.4), and top Open LLM Leaderboard models (§4.5). Further, we conduct an ablation study in a simulated environment in App. B to validate several design choices of CONSTAT.

## 4.1   Experimental Setup

**Reference Models**   We select 20 models from reputable providers, including Meta's LLAMA model families [2, 43], Microsoft's PHI-2 [27] and PHI-3 [1], Google's GEMMA-1.1 [21], several MISTRAL models [28], FALCON-7b [3], and the fully open-source OLMO [25]. A detailed overview of these reference models is available in App. C.

**Benchmarks**   We select a diverse set of four of the most popular LLM benchmarks to evaluate CONSTAT: GSM8k [16] is a benchmark for mathematical reasoning, ARC-Challenge [15] is a multiple-choice benchmark for science questions, MMLU [26] is a multiple-choice general purpose benchmark and Hellaswag [54] is a dataset for commonsense natural language inference. Due to computational constraints, we limit the number of samples in each benchmark to 2000.

**Reference Benchmarks**   To generate reference data for syntax-specific and sample-specific contamination we query GPT-4-TURBO [36] to rephrase samples from the original benchmark and generate new synthetic samples. We generate around 1000 synthetic samples per benchmark and refer to App. C for further details on the generation process. To detect benchmark-specific contamination, we select appropriate reference benchmarks that measure performance on the same task: for GSM8k, we use MathQA [4], for ARC-Challenge we use SCIQ [47], and for Hellaswag we use the Lambada-OpenAI benchmark [38]. For MMLU, we did not select any reference benchmark and thus measured only syntax- and sample-specific contamination.

**Evaluation**   For evaluation, we use the LM Evaluation Harness [20] in a 5-shot setting. We report estimated effects $\hat{\delta}$ along with the p-value for the null hypothesis that the effect $\delta$ is less than 0.

## 4.2 Validating Contamination Detection with CONSTAT in a Controlled Setting

In this section, we demonstrate the effectiveness of CONSTAT in detecting and quantifying contamination in a controlled setting and compare it to multiple baselines. For this purpose, we finetune both LLAMA-2-INSTRUCT-7b and PHI-2 using a variety of hyperparameters and contamination scenarios on each benchmark separately. We vary the number of epochs, the learning rate, the portion of contaminated training samples, whether or not few-shot examples are used during fine-tuning, and whether the model is trained on the original benchmark samples or on rephrased data. For more details, we refer to App. C. We trained a total of 70 models, 9 of which finished training at a loss spike and were therefore excluded from further analysis. 46 of the remaining models were trained on the actual benchmark and should therefore exhibit both syntax- and sample-specific contamination. The rest were trained on rephrased benchmark data and should therefore only exhibit sample-specific contamination. To quantify the true sample-specific contamination effect, we only use half of each benchmark for contamination and measure the performance gap to the other half.

**Detecting Contamination** We first check whether CONSTAT can accurately detect the presence of contamination. We compare CONSTAT against several baselines [12, 35, 40, 41, 52] that aim to detect contamination based on the presence of benchmark samples in the training data. Most of these baselines [12, 35, 41, 52] require a detection threshold to be chosen for each model and benchmark separately. This tuning process requires uncontaminated samples, making it impossible to apply these methods in practice. For comparison to CONSTAT, we tuned these thresholds on the uncontaminated half of the benchmark, which is the most ideal (but unrealistic) scenario. We extract a p-value for these baselines by bootstrapping the samples in the benchmarks and checking how often TPR@1%FPR is bigger than $1\%$. Models are considered contaminated for any method if $p < 0.05$. The only baseline applicable in a realistic setting is Shi [40] and we use their recommendation to consider a model contaminated if the score returned by their method is above $0.85$.

Results in Table 1 show that CONSTAT significantly outperforms all other methods without needing prior knowledge of uncontaminated samples. In particular, we find that CONSTAT can detect $89\%$ of syntax-specifically contaminated models, while the best baseline achieves only $85\%$. The gap widens further for sample-specific contamination, where CONSTAT detects $98\%$ of contaminated models, while the best baseline only detects $71\%$. The only baseline that can be applied in a realistic setting, Shi [40], performs significantly worse than CONSTAT.

Table 1: Percentage of syntax- and sample-specific contaminated models detected by several methods.

| Method | Syntax [%] | Sample [%] |
|---|---|---|
| Carlini et al. [12] | 76.1* | 65.6* |
| Mireshghallah et al. [35] | 76.1* | 68.9* |
| Yeom et al. [52] | 78.3* | 67.2* |
| Shi et al. [41] | 84.8* | 70.5* |
| Shi [40] | 21.7 | 16.4 |
| CONSTAT | **89.1** | **98.4** |

\* indicates that the method needs unrealistic access to uncontaminated samples for hyperparameter selection.

We thus conclude that CONSTAT is the only contamination detection method that can reliably detect contamination and significantly outperforms all baselines even if they are tuned optimally using oracle access to the uncontaminated samples.

**Quantifying Contamination** To evaluate CONSTAT's ability to estimate the sample-specific contamination effect, we compare its estimate to ground truth measurements on uncontaminated samples. As shown in Fig. 2, we observe excellent predictiveness at a coefficient of determination of $r^2 = 0.94$. The only three models that show a significantly higher estimate than the true effect achieve a perfect score on the contaminated samples, capping the true effect and explaining the overestimation.

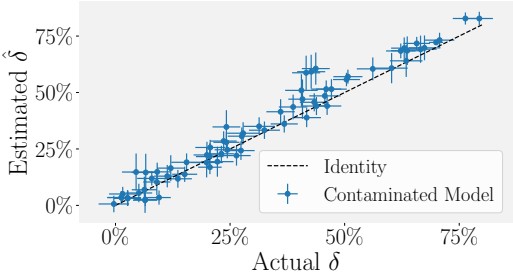

Figure 2: Estimated $\hat{\delta}$ as a function of the true $\delta$ for the finetuned models. 2-sigma intervals are shown.

**Detailed Analysis on GSM8k** We conduct an in-depth analysis of contaminated models finetuned on GSM8k, referring to App. A.2 for a detailed table with all p-values. We finetuned 18 models on this benchmark, one of which remained undetected under sample-specific contamination detection.

Table 2: Contamination results for the reference models on syntax-specific, sample-specific, and benchmark-specific contamination. We only report tests for which the multiple testing corrected p-value is lower than $5\%$ and include the non-corrected p-value, the estimated effect $\hat{\delta}$, the $95\%$ lower bound of the effect $\hat{\delta}_{0.95}$ and the model performance on the benchmark. S stands for sample-specific and B for benchmark-specific contamination. All numbers are reported in percentages.

| Model | Benchmark | Type | Perf. [%] | $p$ [%] | $\hat{\delta}$ [%] | $\hat{\delta}_{0.95}$ [%] |
|---|---|---|---|---|---|---|
| LLAMA-3-70b | ARC | S | 69.03 | 0.03 | 6.61 | 3.21 |
| MISTRAL-7b-v0.1 | GSM8k | S | 39.04 | 0.15 | 8.25 | 4.48 |
| MISTRAL-7b-v0.1 | Hellaswag | S | 83.65 | 0.24 | 3.14 | 1.27 |
| LLAMA-2-INSTRUCT-70b | Hellaswag | S | 85.55 | 0.41 | 3.37 | 1.29 |
| MISTRAL-INSTRUCT-7b-v0.2 | ARC | B | 62.46 | 0.04 | 10.62 | 5.95 |
| MISTRAL-INSTRUCT-7b-v0.2 | Hellaswag | B | 84.55 | 0.18 | 3.52 | 1.56 |
| PHI-2 | GSM8k | B | 58.91 | $< 10^{-2}$ | 36.42 | 26.46 |
| PHI-3-MINI | GSM8k | B | 76.65 | 0.29 | 16.30 | 6.33 |
| OLMO-INSTRUCT-7b | GSM8k | B | 11.75 | $< 10^{-2}$ | 8.86 | 4.99 |

For the detected syntax-specifically contaminated models, we observe an average increase in $\hat{\delta}$ with a factor of 2.28 when transitioning from syntax-specific to sample-specific contamination. This indicates that the models still generalize somewhat to semantically equivalent samples. Furthermore, the models that were not detected by the syntax-specific contamination detection are exactly those models that were trained on rephrased data or were trained for just one epoch. This indicates that these models can still generalize to semantically equivalent samples. Since these scenarios are also more likely to occur in practice, this shows that it is crucial to also consider sample-specific contamination when applying CONSTAT. Finally, the model that remained undetected by the sample-specific contamination detection was a PHI-2 model trained with a lower learning rate. For this model, the actual contamination effect is approximately $5\%$, which is relatively small and thus indicates that CONSTAT is not missing any major contamination.

## 4.3 Contamination of Reputable Reference Models

To determine if our set of reference models exhibit signs of contamination, we perform a leave-one-out analysis, where we evaluate the contamination of model $M$ using $\tilde{\mathcal{M}}_{\mathrm{ref}} \setminus \{M\}$ as reference models. To control for performing multiple p-value tests and reduce the chance of false positives, we apply the Benjamini-Hochberg [9] procedure per benchmark and contamination type to control the false discovery rate at $5\%$. We report all significant results in Table 2 and we discuss them for each type of contamination below.

**Syntax-Specific Contamination**    As expected, we do not find syntax-specific contamination in any reference model, i.e., none of the models fail to generalize to semantically equivalent samples.

**Sample-Specific Contamination**    We find four instances of sample-specific contamination, all with very significant p-values of less than $p = 0.5\%$ and considerable estimated contamination effects between $3\%$ and $8\%$. Specifically, we find contamination of LLAMA-3-70b on ARC, of MISTRAL-7b-v0.1 and LLAMA-2-INSTRUCT-70b on Hellaswag, and MISTRAL-7b-v0.1 on GSM8k. We note that the contamination of LLAMA-2-INSTRUCT-70b on Hellaswag is noted by its model provider [43], but the other model providers do not provide any contamination report for their models.

We investigate these models further on the other benchmarks where the corrected p-value using the Benjamini-Hochberg procedure was not significant. We discuss these results below and refer to App. A for a full overview of their sample-specific contamination. We find that MISTRAL-7b-v0.1 achieves relatively low p-values on both remaining benchmarks ($8\%$ for ARC, $15\%$ for MMLU). Furthermore, we additionally evaluated MISTRAL-7b-v0.2 after obtaining these results and found similar results for this model (see Table 17 in App. E). Therefore, we exclude MISTRAL-7b-v0.1 from our set of reference models. While in particular LLAMA-3-70b also exhibits low p-values for other benchmarks, none fall below $p \leqslant 1\%$. It is thus highly likely that also LLAMA-3-70b and LLAMA-2-INSTRUCT-70b are contaminated across several benchmarks, but we keep both as reference models to ensure that we do not obtain a higher false positive rate in our further analysis.

**Benchmark-Specific Contamination** While we find several instances of benchmark-specific contamination in the reference models, several at very low p-values ($p < 0.01\%$), this requires a more nuanced interpretation. For example, both PHI models exhibit very large effect sizes ($> 15\%$) and small p-values ($p < 0.01\%$) for contamination on GSM8k. We suspect that this is due to their reasoning-focused training process and small model size. While GSM8k allows free text answers, giving the model tokens to reason, MathQA is a multiple-choice benchmark that requires the model to answer with a single token indicating the chosen option and therefore gives no room for this reasoning ability to shine.

### 4.4 Contamination of Popular Model Families

We now use CONSTAT to detect contamination in four popular model families, discussing results for QWEN-1.5 [6] and YI [53] below, while deferring discussions of STABLELM-2 [8] and INTERNLM-2 [11] to App. A.1.

**QWEN-1.5** We evaluate all chat models from the QWEN-1.5 model family, with sizes 1.8b, 4b, 7b, 14b, 72b, and 110b. The only case of sample-specific contamination is for the 4b model on GSM8k with $p < 10^{-4}$ and an estimated effect of $5.4\%$. The larger models show significant benchmark-specific contamination on ARC and Hellaswag, with p-values smaller than $1\%$ and estimated effects between $8\%$ and $14\%$.

**YI** We evaluate both the 6b and 34b parameter base models of the YI model-family. Only YI-34b shows significant contamination, with sample-specific contamination at $p < 0.2\%$ and estimated effects of around $6\%$ on both ARC and Hellaswag. We find additional sample-specific contamination on GSM8k of around $4\%$ at a p-value of $p = 6\%$ and *syntax-specific* contamination on Hellaswag at a p-value of $p = 5\%$. Thus, we conclude that this model shows significant contamination across multiple benchmarks.

### 4.5 Contamination of Top Open LLM Leaderboard Models

We use CONSTAT to investigate contamination in the top three 7B models on the open LLM Leaderboard[2], BARRAHOME/MISTROLL-7b-v2.2, YAM-PELEG/EXPERIMENT26-7b, and MTSAIR/-MULTI_VERSE_MODEL and find that all three models exhibit significant benchmark-specific contamination. Specifically, all models show strong contamination with estimated effects of $\hat{\delta} > 10\%$ for the benchmarks where the reference benchmark is not included in the Open LLM Leaderboard (GSM8k, Hellaswag, and ARC). Further, all models show significant sample-specific contamination on GSM8k with $\hat{\delta} \approx 9\%$. For more detailed results, we refer to App. A.

This inflated performance could be caused by a model selection bias, as the Open LLM Leaderboard features thousands of models. This issue is exacerbated by the recent trend of merging models [22, 50] where hyperparameters are frequently selected based on their benchmark performance. We therefore urge the community to be more cautious when selecting models from the leaderboard.

## 5 Related Work

**Contamination Detection** Contamination detection methods can be broadly divided into two main categories. The first category [10, 14, 19, 29, 36, 43, 45, 51] focuses on analyzing the training data directly to identify overlaps with the benchmarks used for model evaluation. However, training data is rarely shared, even for open-weight models, making it irrelevant for third-party contamination detection. The second category [18, 23, 24, 31, 34, 37, 40, 41, 49] relies solely on access to the model and its predictions, aiming to detect contamination through model queries. As noted by Dekoninck et al. [17], some of these methods require metadata (e.g., benchmark name, canonical ordering) to be leaked along with the benchmark samples in the training data [23, 24, 37]. Methods that do not require metadata depend on perplexity-based metrics to measure the model's uncertainty on benchmark samples, but these can be easily circumvented by training on rephrased samples [17]. It is important to note that none of these methods can estimate the influence of contamination and that they are outperformed by CONSTAT in terms of detection accuracy (see §4.2).

---

[2]Rank on Open LLM Leaderboard as of the 4th of May 2024.

An alternative approach is presented by Zhu et al. [57], who measure model performance on rephrased benchmarks instead of the original benchmarks to obtain more accurate estimates of model performance. However, their results vary significantly across benchmarks, they do not provide a statistical framework for contamination detection, and they only demonstrate that evaluating on rephrased samples *partially* recovers the results of uncontaminated base models. Furthermore, they do not go beyond measuring performance on rephrased benchmarks and can therefore also be evaded by training on rephrased samples [17].

**Reference Benchmarks**   Recent studies have introduced new benchmarks designed to evaluate performance on tasks similar to those in prior popular benchmarks and thus can be used to estimate the degree of contamination. GSM1k [55] was developed to closely replicate the efforts behind GSM8k and to compare model performances between these benchmarks. However, GSM1k lacks a statistical test, and the slight variations between GSM8k and GSM1k might partially explain the contamination levels observed in their analysis. Another recent benchmark, SWE-bench [30], focuses on evaluating performance on coding tasks. By comparing their results with those of Human-Eval [13], one can visually interpret potential contamination in Human-Eval. However, the absence of a statistical test hinders precise contamination detection. In both scenarios, CONSTAT can improve their findings, enabling accurate estimations of contamination in existing models.

# 6   Discussion

**Limitations**   Our method estimates the effect of contamination on performance relative to a set of reference models. Therefore, if these reference models are also contaminated, our method only measures the effect relative to this base level of contamination. However, our leave-one-out experiment, presented in §4.3, helps identify and exclude contaminated models, partially mitigating this limitation. Furthermore, it is important to note that accurate relative performance measurements are sufficient for both model selection and to assess methodological improvements, which are the most important use cases of benchmarks.

Further, our work uses an LLM to generate synthetic samples, introducing potential distributional biases into the synthetic benchmark $D_{\text{ref}}$. We briefly discuss these biases here. Firstly, synthetic benchmark may contain more mislabeled samples. However, since these samples equally affect all models, CONSTAT accounts for this in its difficulty correction. Secondly, synthetic samples generated by a model are likely easier for that model itself to solve. Therefore, contamination results for the model used to generate the samples would be unreliable for sample-specific contamination detection. However, these limitations are not inherent flaws of CONSTAT, and can be mitigated by using more sophisticated synthetic benchmark generation techniques.

**Impact**   Model evaluation is a crucial part of LLM development, with benchmarks playing a key role in evaluating model performance on tasks like code generation, question answering, and summarization. Contamination of these benchmarks can inflate performance estimates, potentially misleading researchers and practitioners. To address this, CONSTAT provides a statistical framework to estimate the impact of contamination on model performance. This enables more accurate evaluations and allows for the removal of suspicious models from leaderboards, ensuring a fairer evaluation of model capabilities. Furthermore, it is important to note that CONSTAT can be applied to any model, not just LLMs, as long as the model's performance can be measured on a benchmark.

# 7   Conclusion

We present CONSTAT, a statistical framework designed to detect contamination and estimate its effect on model performance. Unlike existing methods, CONSTAT is based on a novel, performance-based definition of contamination and compares performance with various reference benchmarks to obtain a detailed contamination analysis that distinguishes between syntax-, sample-, and benchmark-specific contamination. We investigate CONSTAT's effectiveness in an extensive control study and demonstrate that it not only outperforms existing methods but also, in contrast to them, does not require prior knowledge about uncontaminated samples. Finally, we use CONSTAT to investigate contamination in popular models and find, among others, very high levels of contamination in MISTRAL-7b-v0.1 and YI-34b and high levels of contamination in LLAMA-3-70b and LLAMA-2-INSTRUCT-70b.

## Acknowledgements

This work has been done as part of the EU grant ELSA (European Lighthouse on Secure and Safe AI, grant agreement no. 101070617) and was funded in part by the Swiss National Science Foundation (SNSF) [200021_207967]. Views and opinions expressed are however those of the authors only and do not necessarily reflect those of the European Union or European Commission. Neither the European Union nor the European Commission can be held responsible for them.

The work has received funding from the Swiss State Secretariat for Education, Research and Innovation (SERI).

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

Table 3: Full overview of sample-specific contamination in MISTRAL-7b-v0.1, LLAMA-2-INSTRUCT-70b and LLAMA-3-70b. All numbers are reported in percentages.

| Model | Benchmark | Perf. [%] | $p$ [%] | $\hat{\delta}$ [%] | $\hat{\delta}_{0.95}$ [%] |
|---|---|---|---|---|---|
| LLAMA-2-INSTRUCT-70b | ARC | 61.86 | 14.68 | 1.96 | −1.01 |
| | GSM8k | 55.80 | 58.60 | −0.41 | −4.42 |
| | Hellaswag | 85.55 | 0.41 | 3.37 | 1.29 |
| | MMLU | 56.85 | 32.91 | 0.71 | −2.10 |
| LLAMA-3-70b | ARC | 69.03 | 0.03 | 6.61 | 3.21 |
| | GSM8k | 81.58 | 15.45 | 2.05 | −1.49 |
| | Hellaswag | 86.45 | 1.01 | 2.86 | 0.90 |
| | MMLU | 76.46 | 5.76 | 3.35 | −0.21 |
| MISTRAL-7b-v0.1 | ARC | 58.96 | 7.91 | 2.21 | −0.40 |
| | GSM8k | 39.04 | 0.15 | 8.25 | 4.48 |
| | Hellaswag | 83.65 | 0.24 | 3.14 | 1.27 |
| | MMLU | 58.01 | 15.23 | 1.88 | −1.05 |

# A  Additional Results

We present the complete results for the experiments discussed in §4 here and include a discussion on the STABLELM-2 and INTERNLM-2 model families. We provide a table with the results for all evaluated model families where $p < 1\%$ in Table 4.

## A.1  Discussion on INTERNLM-2 and STABLELM-2

**INTERNLM-2**  We evaluated four models in the INTERNLM-2 model family: the models with size 1.8b and 7b, and the math-base and math models, also of 7b parameters. Overall, we found very little evidence of contamination in these models, with no model showing significant ($p < 1\%$) sample-specific contamination. However, we did find some evidence for benchmark-specific contamination for GSM8k and Hellaswag for several models in the model family. Specifically, INTERNLM-2-7b and INTERNLM-2-MATH-7b show significant benchmark-specific contamination on GSM8k with $p < 0.5\%$ and estimated effects of $20\%$ and $40\%$ respectively. The size of this effect is likely due to the same reasons as the measured contamination in the PHI models, where the models are too small to solve mathematical questions in one go and have been trained/finetuned to perform chain-of-thought mathematics. The benchmark-specific contamination on Hellaswag is present for all three 7b models in the family, with $0.5\% < p < 1\%$ and estimated effects of $6\%$ to $11\%$.

**STABLELM-2**  For STABLELM-2, we evaluated 6 models in the model family (12b, INSTRUCT-12b, 1.6b, INSTRUCT-1.6b, ZEPHYR-3b and ALPHA-7b-v2). We found only one instance of sample-specific contamination, with the 12b model showing slight ($p = 0.7\%$) contamination for ARC. Notably, we found that all of the models in this family show benchmark-specific contamination for GSM8k with estimated effects between $15\%$ and $40\%$.

## A.2  Results for GSM8k Contaminated Models

We present the complete results for the contaminated models finetuned on the GSM8k benchmark in Table 5. For a detailed explanation of each of the settings, we refer to App. C. We do mention here that in the realistic setting, we only train for 1 epoch, without any few-shot samples in the prompt and with additional background instruction-tuning data from the OpenOrca dataset [32].

Table 4: Complete results for all evaluated model families for all tests with result $p < 1\%$. All numbers in the table are reported in percentages.

| Model | Benchmark | Type | Perf. [%] | $p$ [%] | $\hat{\delta}$ [%] | $\hat{\delta}_{0.95}$ [%] |
|---|---|---|---|---|---|---|
| QWEN-INSTRUCT-1.5-14b | ARC | B | 56.91 | 0.71 | 11.77 | 5.94 |
| QWEN-INSTRUCT-1.5-72b | ARC | B | 64.68 | 0.01 | 12.59 | 7.61 |
| QWEN-INSTRUCT-1.5-110b | ARC | B | 69.45 | 0.12 | 9.33 | 4.47 |
| QWEN-INSTRUCT-1.5-4b | GSM8k | S | 6.52 | $< 10^{-2}$ | 5.35 | 4.01 |
| QWEN-INSTRUCT-1.5-7b | Hellaswag | B | 78.65 | 0.74 | 7.46 | 3.00 |
| QWEN-INSTRUCT-1.5-14b | Hellaswag | B | 82.15 | 0.07 | 6.48 | 3.74 |
| QWEN-INSTRUCT-1.5-72b | Hellaswag | B | 86.35 | $< 10^{-2}$ | 8.24 | 6.05 |
| YI-34b | ARC | S | 63.99 | 0.20 | 5.00 | 2.12 |
| YI-34b | Hellaswag | S | 86.15 | $< 10^{-2}$ | 6.51 | 4.40 |
| YI-34b | Hellaswag | B | 86.15 | 0.14 | 3.96 | 1.89 |
| INTERNLM-2-7b | GSM8k | B | 62.09 | 0.43 | 19.27 | 7.98 |
| INTERNLM-2-MATH-7b | GSM8k | B | 72.93 | $< 10^{-2}$ | 39.40 | 27.15 |
| INTERNLM-2-7b | Hellaswag | B | 80.10 | 0.41 | 6.58 | 3.19 |
| INTERNLM-2-MATH-7b | Hellaswag | B | 77.65 | 0.90 | 8.55 | 3.11 |
| INTERNLM-2-MATH-BASE-7b | Hellaswag | B | 79.65 | 0.40 | 11.41 | 5.51 |
| STABLELM-2-12b | ARC | S | 59.47 | 0.68 | 4.61 | 1.59 |
| STABLELM-2-1.6b | GSM8k | B | 18.88 | $< 10^{-2}$ | 16.56 | 12.95 |
| STABLELM-2-INSTRUCT-1.6b | GSM8k | B | 42.00 | $< 10^{-2}$ | 27.79 | 19.27 |
| STABLELM-2-ZEPHYR-3b | GSM8k | B | 51.63 | $< 10^{-2}$ | 48.78 | 44.34 |
| STABLELM-2-12b | GSM8k | B | 58.00 | 0.39 | 17.92 | 6.96 |
| STABLELM-2-INSTRUCT-12b | GSM8k | B | 68.84 | $< 10^{-2}$ | 32.93 | 21.09 |
| STABLELM-2-INSTRUCT-12b | Hellaswag | B | 86.25 | $< 10^{-2}$ | 7.04 | 4.88 |
| yam-peleg/EXPERIMENT26-7b | ARC | B | 72.44 | $< 10^{-2}$ | 22.36 | 17.28 |
|  | GSM8k | S | 74.53 | 0.24 | 7.60 | 3.19 |
|  | GSM8k | B | 74.53 | $< 10^{-2}$ | 29.69 | 18.15 |
|  | Hellaswag | B | 88.60 | $< 10^{-2}$ | 13.11 | 10.27 |
| BARRAHOME/MISTROLL-7b-v2.2 | ARC | B | 72.53 | $< 10^{-2}$ | 22.21 | 17.06 |
|  | GSM8k | S | 74.53 | 0.34 | 7.34 | 3.05 |
|  | GSM8k | B | 74.53 | $< 10^{-2}$ | 29.28 | 17.77 |
|  | Hellaswag | B | 88.60 | $< 10^{-2}$ | 12.95 | 10.15 |
| MTSAIR/multi_verse_model | ARC | B | 72.44 | $< 10^{-2}$ | 22.12 | 16.97 |
|  | GSM8k | S | 74.68 | 0.53 | 7.20 | 2.65 |
|  | GSM8k | B | 74.68 | $< 10^{-2}$ | 29.84 | 18.43 |
|  | Hellaswag | B | 88.55 | $< 10^{-2}$ | 12.14 | 9.56 |

Table 5: Complete results for the contaminated models finetuned on GSM8k. LLAMA-2 is the LLAMA-2-INSTRUCT-7b model. $\delta$ is the actual effect measured on the uncontaminated samples. The other values are the estimates p-values and effects for syntax- and sample-specific contamination. All numbers in the table are reported in percentages.

| Model | Setting | Perf. [%] | $\delta$ [%] | $p_{\text{syntax}}$ [%] | $\hat{\delta}_{\text{syntax}}$ [%] | $p_{\text{sample}}$ [%] | $\hat{\delta}_{\text{sample}}$ [%] |
|---|---|---|---|---|---|---|---|
| LLAMA-2 | Default | 92.11 | 79.38 | $< 10^{-2}$ | 40.35 | $< 10^{-2}$ | 82.77 |
| | Default, rephrased | 64.64 | 50.40 | 99.14 | $-6.80$ | $< 10^{-2}$ | 55.72 |
| | learning rate $10^{-4}$ | 73.29 | 69.96 | $< 10^{-2}$ | 43.60 | $< 10^{-2}$ | 72.18 |
| | learning rate $10^{-5}$ | 38.85 | 15.51 | $< 10^{-2}$ | 12.62 | $< 10^{-2}$ | 19.08 |
| | Trained for 1 epoch | 25.19 | 7.92 | 0.94 | 5.42 | $< 10^{-2}$ | 11.94 |
| | Other few-shot samples | 89.53 | 76.35 | $< 10^{-2}$ | 38.08 | $< 10^{-2}$ | 82.80 |
| | No few-shot samples | 80.27 | 65.73 | $< 10^{-2}$ | 37.36 | $< 10^{-2}$ | 71.74 |
| | Realistic | 69.80 | 50.71 | $< 10^{-2}$ | 32.38 | $< 10^{-2}$ | 56.99 |
| | Realistic, rephrased | 40.52 | 20.67 | 94.50 | $-4.47$ | $< 10^{-2}$ | 25.65 |
| PHI-2 | Default | 79.51 | 36.03 | $< 10^{-2}$ | 14.96 | $< 10^{-2}$ | 41.46 |
| | Default, rephrased | 69.20 | 19.95 | 94.93 | $-4.07$ | $< 10^{-2}$ | 22.31 |
| | learning rate $10^{-4}$ | 82.25 | 62.25 | $< 10^{-2}$ | 33.39 | $< 10^{-2}$ | 68.42 |
| | learning rate $10^{-5}$ | 60.09 | 6.45 | 39.06 | 0.64 | 21.38 | 2.27 |
| | Trained for 1 epoch | 55.39 | 9.02 | 6.98 | 3.90 | 0.05 | 10.34 |
| | Other few-shot samples | 81.34 | 38.76 | $< 10^{-2}$ | 19.95 | $< 10^{-2}$ | 43.56 |
| | No few-shot samples | 64.19 | 20.55 | 0.75 | 6.48 | $< 10^{-2}$ | 21.61 |
| | Realistic | 59.79 | 12.06 | 4.93 | 4.30 | $< 10^{-2}$ | 16.45 |
| | Realistic, rephrased | 60.55 | 6.30 | 88.04 | $-2.73$ | 1.48 | 6.92 |

# B  Ablation Study via Simulation

To further investigate the performance of CONSTAT, we conduct an ablation study using simulations. This approach allows us to test various scenarios and understand the behavior of CONSTAT under different conditions without the need for finetuning or computationally intense evaluations. Furthermore, it helps in verifying the p-values returned by various tests while avoiding the risk of tuning our final test to our analysis in §4. We first explain the simulation setup and then present the results.

## B.1  Simulation and Setup

**Simulation**  In the simulation, samples are modeled as real numbers representing their complexity. Each sample $x$ in a benchmark $D$ is drawn from a benchmark-specific distribution $\mathcal{D}$. Therefore, a benchmark $D$ can be specified by a number $n \in \mathbb{Z}_{>0}$, indicating the number of samples, and a distribution $\mathcal{D}$, from which the samples are drawn. Given the benchmarks $(n, \mathcal{D})$ and $(n_{\text{ref}}, \mathcal{D}_{\text{ref}})$, a model is represented as $M = (m, m_{\text{ref}}) \in \mathbb{R}^2$, where each number indicates the quality of the model on the respective benchmarks. The probability that a model $M$ answers a sample $x \in D$ correctly is given by the formula $\min(1, \exp(-x/m))$. Note that this probability increases as the quality of the model $m$ grows and decreases as the complexity of the sample $x$ increases.

If $m_{\text{ref}} < m$ for a given model, the model is contaminated. Reference models can be drawn from a distribution $\mathcal{M}$ over the real numbers such that $m_{\text{ref}} = m$ for each reference model. To simulate noise in the evaluation of models, we can add noise to the quality of the reference models, resulting in $m_{\text{ref}} \approx m$.

**Statistical Tests**  We compare CONSTAT against various other statistical tests that one could construct. First, we include various variants of CONSTAT. CONSTAT-NO-SORT does not sort the reference models by their and fits the hardness correction function $H_{D_{\text{ref}}}$ directly on the scores $(S_{D_{\text{ref}}}(M_i), S_D(M_i))$. CONSTAT-NO-RANDOM does not include a random model in the set of reference models. CONSTAT-NO-BOOTSTRAP only performs bootstrapping over the samples and not over the reference models.

We also include two alternative tests. MEAN-TEST directly compares performance on the reference and original benchmarks, considering a model contaminated if its performance on the original benchmark is significantly higher. NORMALIZED-TEST instead computes the normalized performance with respect to the models by computing the means $\mu_D, \mu_{D_{\text{ref}}}$ and standard deviations $\sigma_D, \sigma_{D_{\text{ref}}}$ of the reference models on each benchmark. It then bootstraps the reference models and samples to obtain the p-value as the probability that the normalized performance $\sigma_D^{-1}(S_D(M) - \mu_D)$ of the model on the original benchmark is higher than on the reference benchmark. Therefore, NORMALIZED-TEST essentially corrects for first- and second-order distributional differences between $\mathcal{D}$ and $\mathcal{D}_{\text{ref}}$.

**Reporting Results**  We report results for specific distributions $\mathcal{M}, \mathcal{D}$ and $\mathcal{D}_{\text{ref}}$ for a model $M$ for which we aim to detect possible contamination. For each choice of distributions, we run 1000 simulations, each drawing new reference models and benchmarks from the given distributions and performing the tests described before. This ablation focuses on the uncontaminated case, where $m = m_{\text{ref}}$, as avoiding false positives is crucial. In Fig. 3a, we show an example plot of the resulting CDF of the returned p-value when $\mathcal{D} = \mathcal{D}_{\text{ref}}$. As expected, the CDF for each method is very close to the identity line for the uncontaminated model. Ideally, the curve for uncontaminated models should be as close as possible to this identity line to ensure reliable p-values. Above the line is especially problematic, as this would be the cause of false positives. Furthermore, by swapping the distributions $\mathcal{D}$ and $\mathcal{D}_{\text{ref}}$, one would obtain a mirror image of the plot. This means that a CDF that is below the identity line in a given situation, would be above the identity line in the mirror image, and is therefore also problematic for the same reason.

We now report results for various scenarios. In each case, we aim to ensure that a specific test fails and explain why this is the case. In all explored scenarios, CONSTAT performs as expected, while the other methods fail. Unless specified otherwise, we use 20 reference models and 1000 samples for each benchmark, in line with our results presented in §4. A full overview of each parameter setting can be found in Table 6. We always set the quality of the model under consideration to $(1, 1)$.

Table 6: Settings for the simulation scenarios. We use the union of two distributions to indicate the distribution that samples from both distributions with equal probability. The $\sigma$ column indicates the noise added to the reference models, using a normal distribution with mean 0 and standard deviation $\sigma$. A normal distribution is denoted using the notation $\mathcal{N}(\mu, \sigma)$ where $\sigma$ is the standard deviation.

| Scenario | $\mathcal{M}$ | $\mathcal{D}$ | $\mathcal{D}_{\text{ref}}$ | $\sigma$ |
|---|---|---|---|---|
| Different Distributions | $\mathcal{N}(1, 0.3)$ | $\mathcal{N}(0.4, 0.3)$ | $\mathcal{N}(0.8, 0.2)$ | 0 |
| Non-Linearity | $\mathcal{N}(0.6, 0.2)$ | $\mathcal{N}(0.8, 0.1) \cup \mathcal{N}(1.4, 0.1)$ | $\mathcal{N}(0.3, 0.1) \cup \mathcal{N}(1, 0.1)$ | 0 |
| Noise | $\mathcal{N}(0.8, 0.1)$ | $\mathcal{N}(1, 0.4)$ | $\mathcal{N}(1, 0.4)$ | 0.05 |
| Bootstrapping Models | $\mathcal{N}(0.6, 1)$ | $\mathcal{N}(0.8, 0.1) \cup \mathcal{N}(1.4, 0.1)$ | $\mathcal{N}(0.3, 0.1) \cup \mathcal{N}(1, 0.1)$ | 0.1 |
| No Random Model | $\mathcal{N}(4, 1)$ | $\mathcal{N}(4, 0.2) \cup \mathcal{N}(0.8, 0.8)$ | $\mathcal{N}(0.8, 0.8)$ | 0.05 |

## B.2 Results

**Different Distributions**  The MEAN-TEST should fail if the difficulty of one benchmark is different than the other. We slightly decrease the difficulty of the samples in the original benchmark to make MEAN-TEST return false positives, as shown in Fig. 3b. Despite $M$ being uncontaminated, the p-values returned by MEAN-TEST show a very steep CDF.

**Non-Linearity**  NORMALIZED-TEST assumes a linear relationship between performances on reference and original benchmarks, but non-linear relationships can occur. For instance, it is non-linear for sample-specific contamination in the GSM8k benchmark (see Fig. 1). Therefore, we change the benchmark distributions to ensure a non-linear relationship. The result shown in Fig. 3c shows that NORMALIZED-TEST returns a very steep CDF. We note that CONSTAT-NO-SORT also returns a steep CDF in this particular case.

**Noise**  When reference models do not have the same quality on both benchmarks, noise is introduced in the signal that the test receives. Our theoretical analysis in §3 corrects for this noise by sorting the reference models by performance on each benchmark. CONSTAT-NO-SORT is more susceptible to this noise. We showcase this for an uncontaminated model in Fig. 3d by keeping $\mathcal{D} = \mathcal{D}_{\text{ref}}$, but now adding a small amount of noise to the reference models. CONSTAT-NO-SORT returns a steep CDF and should not be used in practice due to the noisy nature of real-world scenarios.

**Bootstrapping Models**  Bootstrapping over reference models is necessary for reliable p-values. Without bootstrapping, the test would rely on the specific instantiation of the reference models, leading to p-values that are either too certain, always returning 0 or 1. We repeat the non-linear scenario with added noise to the reference models and a wider distribution over them. As shown in Fig. 3e, CONSTAT-NO-BOOTSTRAP returns a CDF that is very steep at either edge.

**No Random Model**  Adding a random model to the reference models in CONSTAT provides further regularization. The effect of this addition only becomes apparent when we use fewer reference models in a non-linear scenario. In such cases, all reference models are relatively close together and the smoothing spline overfits to this local part of the curve. We demonstrate this in Fig. 3f, using only five reference models and a non-linear relationship between the benchmarks. CONSTAT-NO-RANDOM shows a rather steep CDF in this scenario.

We conclude that CONSTAT is robust to various scenarios and provides reliable p-values in all cases. The other tests fail in the scenarios we have presented, highlighting the importance of the design choices made in CONSTAT.

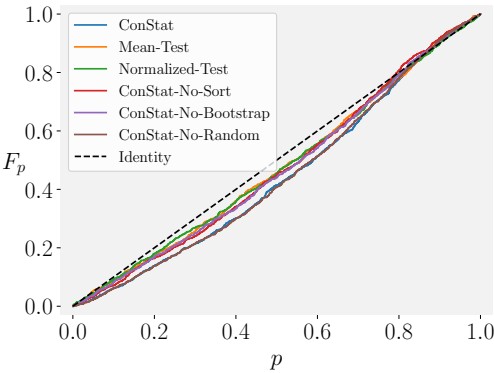
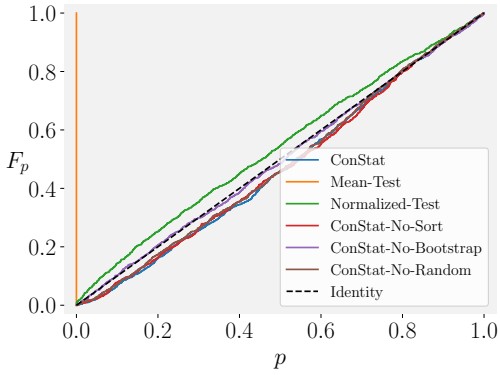

a A simple scenario where all tests should return a CDF close to the identity line.

b A scenario where we make the distributions of the benchmarks different.

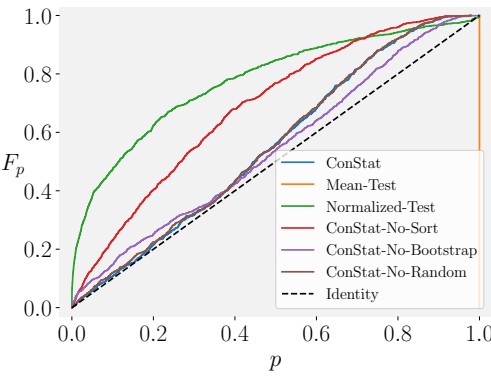
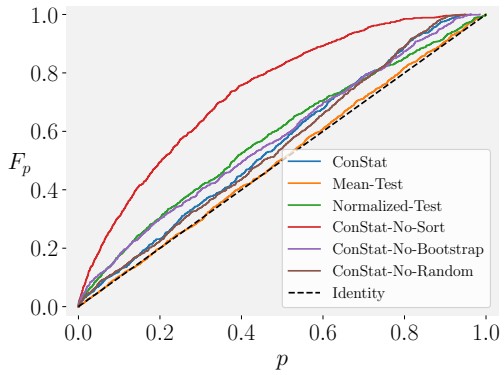

c The relationship between performances on the reference and original benchmarks is non-linear.

d The reference models are noisy and the relationship between the benchmarks is linear.

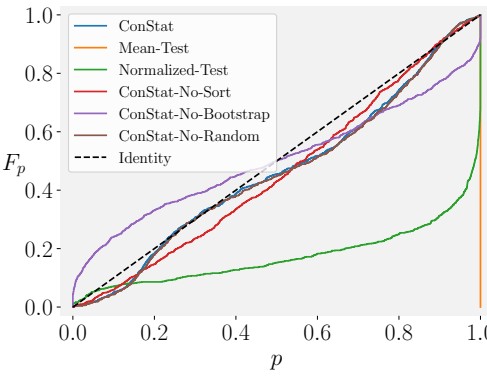
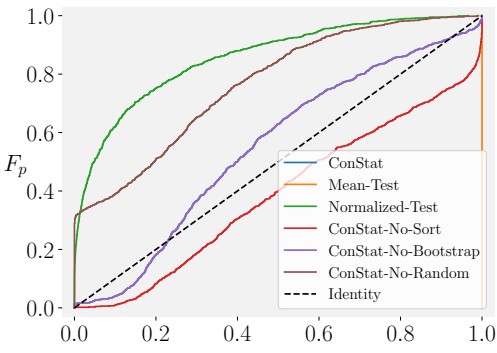

e The reference models are noisy and the relationship between the benchmarks is non-linear.

f A small number of reference models and a non-linear relationship between the benchmarks. CONSTAT-NO-BOOTSTRAP and CONSTAT are the same in this case.

Figure 3: CDF of various statistical tests for uncontaminated models in different scenarios.

# C   Experimental Details

We describe the full details for the experiments presented in §4. This includes the preprocessing stage of the benchmarks, the data generation process, the fine-tuning of the models on the benchmarks, and the evaluation of the models, including the reference models used. Additionally, we provide details on the computational resources necessary to run the experiments. Licensing information for all assets used in the experiments is provided in App. D.

**Preprocessing**   We select four benchmarks for our experiments, ARC [15], GSM8k [16], Hellaswag [54], and MMLU [26]. Due to the large size of MMLU, we first select a subset of the topics from which it consists. Specifically, we select the following topics: Abstract Algebra, Anatomy, Astronomy, Business Ethics, Clinical Knowledge, College Biology, College Chemistry, College Computer Science, College Mathematics, College Medicine, College Physics, Computer Security, Conceptual Physics, Econometrics, and Electrical Engineering. We randomly select 2000 samples from each of the benchmarks. These samples were then split into two equally-sized sets, one of which was used for contaminating the fine-tuned models.

For the chosen reference benchmarks we limit the number of samples to 2000. We choose this number based on the trade-off between tight confidence bounds and computational budget. Computational complexity increases linearly with the number of samples, while the size of confidence intervals decreases. We found that 2000 samples provide tight confidence bounds and allow us to evaluate over 50 models within our budget.

**Data Generation**   For each benchmark, we generate a rephrased version of the benchmark and a synthetic benchmark. For both these purposes, we use GPT-4-TURBO [36]. Specifically, for the rephrased benchmarks, we use a system prompt asking the model to rephrase the input (including options for multiple-choice benchmarks) of a given sample. We use a different system prompt to generate rephrased training samples, including the input and output, to finetune the models trained on rephrased data for our experiments in §4.2. By using separate prompts for training and evaluation, we ensure that the evaluation did not occur on the same data as the training.

For the synthetic benchmarks, we write a system prompt that asks to generate new synthetic samples for the benchmark. To obtain faithful synthetic samples, we use few-shotting where the model is given several examples of the benchmark. By placing these generated samples in the "assistant" field of the chat model and changing the given few-shot examples for each sample we generate, we ensure both faithful and diverse samples. We generate 1000 samples for each benchmark.

To ensure high data quality for rephrasing and synthetic sample generation, we performed the following procedure:

- We manually tested around 10 samples for each benchmark with various system prompts, iteratively refining the prompts until we were satisfied with the output quality.

- We performed a manual check of approximately 100 samples for each benchmark to identify common mistakes and evaluate overall data quality. For instance, for the GSM8k benchmark, we found that some generated samples did not result in an integer answer, or the model used a rounding operation. These samples were removed by checking if the answer was an integer and ensuring no rounding was involved.

Post-processing was then applied to the synthetically generated benchmark samples. First, duplicates within the synthetic samples were removed by searching for high 1-gram overlap ratios between two samples. Second, we removed samples with a high 1-gram overlap ratio with the original benchmark samples, ensuring the synthetic samples were not too similar to the originals.

The system prompts used for the rephrased benchmarks and synthetic benchmarks are available in the code repository.

**Finetuning**   We explain the finetuning process for the PHI-2 and LLAMA-2-INSTRUCT-7b models that were used in §4.2. We use the Hugging Face Transformers library [48] for the finetuning process.

Specifically, we applied full finetuning with batch size 16 and the Adam optimizer on different datasets and using different hyperparameters. We use the following default hyperparameters:

- A learning rate of $5 \cdot 10^{-5}$.
- The dataset on which we train is the contaminatable part of a given benchmark.
- We train for 5 epochs.
- The prompt includes the exact few-shot samples used for evaluation.

We then train 8 other models where we always change specific parameters in this default setting. Specifically, we train models that diverge from the default setting in the following ways:

1. Instead of training with the exact samples from the benchmark, we train on the rephrased benchmark.
2. We change the learning rate to $10^{-5}$.
3. We change the learning rate to $10^{-4}$.
4. We only train for 1 epoch.
5. We train without any few-shot samples in the prompt.
6. We train with a random set of few-shot samples instead of the few-shot samples from the benchmark.
7. We do not include any few-shot samples in the prompt, include additional background instruction-tuning data from the OpenOrca dataset [32], and only train for 1 epoch.
8. We do the same as in the previous setting, with the additional change that we train on the rephrased benchmark instead of the actual one.

By including such a wide range of possible settings, we ensure that we cover a wide range of possible contamination effects. As can be seen in Fig. 2, the resulting models indeed show varying levels of contamination from $0\%$ up to $80\%$.

**Reference Models** The following models were used as reference models in our experiments: PHI-2, PHI-3, LLAMA-2-7b, LLAMA-2-INSTRUCT-7b, LLAMA-2-13b, LLAMA-2-INSTRUCT-13b, LLAMA-2-INSTRUCT-70b, LLAMA-3-8b, LLAMA-3-INSTRUCT-8b, LLAMA-3-70b, LLAMA-3-INSTRUCT-70b, MISTRAL-7b-v0.1, MISTRAL-INSTRUCT-7b-v0.1, MISTRAL-INSTRUCT-7b-v0.2, MIXTRAL-INSTRUCT-8x7b, MIXTRAL-INSTRUCT-8x22b, FALCON-7b, FALCON-INSTRUCT-7b, GEMMA-1.1-7b, GEMMA-1.1-INSTRUCT-7b, OLMO-INSTRUCT-7b. As discussed in §4.3, we removed MISTRAL-7b-v0.1 from the reference models after a contamination analysis.

**Evaluation** We evaluate the models with v0.4.1 of the LM Evaluation Harness [20]. We use 5-shot evaluation for all models and provide the custom fork of the evaluation harness to allow for the evaluation on all the synthetic and rephrased benchmarks in our code repository.

**Compute** We spent around 300 USD on the OpenAI API to generate all benchmarks. Furthermore, we used a single Nvidia H100 GPU for around 1 month to finetune and evaluate all models. Finally, for models that were too large to fit on a single GPU, we used the Together API to run inference. We spent an additional 263 USD on this platform.

# D Licensing Information

We include the license for all models, benchmarks and other assets used in this paper in Table 7.

Table 7: Table with assets used, description of their use and the license under which they are distributed. Sections are split by the type of asset: benchmarks, code repositories and then models.

| Asset | Description & Use | License Name |
|---|---|---|
| MMLU [26] | General-purpose benchmark used for evaluation | MIT License |
| Hellaswag [54] | General-purpose benchmark used for evaluation | MIT License |
| GSM8k [16] | General-purpose benchmark used for evaluation | MIT License |
| ARC-Challenge [15] | General-purpose benchmark used for evaluation | CC-BY-SA-4.0 |
| OpenOrca [32] | Instruction-tuning dataset used in finetuning process | MIT License |
| LM Evaluation Harness [20] | Framework used to perform evaluations | MIT License |
| [40] | Repository to run the [40] baseline | Not Specified |
| Scipy [44] | Adapted code for smoothing spline fitting | BSD 3-Clause License |
| LLAMA-2 [43] | Includes all LLAMA-2 models, were evaluated for contamination and used for finetuning | Llama 2 Community License Agreement |
| LLAMA-3 [2] | Includes all LLAMA-3 models, were evaluated for contamination | Llama 3 Community License Agreement |
| FALCON [3] | Includes all FALCON models, were evaluated for contamination | Apache 2.0 License |
| GEMMA-1.1 [21] | Includes all GEMMA-1.1 models, were evaluated for contamination | Gemma Terms of Use |
| YAM-PELEG/EXPERIMENT26-7b | Was evaluated for contamination | Apache 2.0 License |
| BARRAHOME/MISTROLL-7b-v2.2 | Was evaluated for contamination | MIT License |
| MTSAIR/MULTI_VERSE_MODEL | Was evaluated for contamination | Apache 2.0 License |
| MISTRAL [28] | Includes all MISTRAL models, were evaluated for contamination | Apache 2.0 License |
| PHI-2 [27] | Was evaluated for contamination and used for finetuning | MIT License |
| PHI-3 [1] | Was evaluated for contamination. | MIT License |
| QWEN-1.5 [6] | Includes all QWEN-1.5 models, were evaluated for contamination | Tongyi Qianwen License Agreement |
| STABLELM-2 [8] | Includes all STABLELM-2 models, were evaluated for contamination | Stability Ai Non-Commercial Research Community License Agreement |
| INTERNLM-2 [11] | Includes all INTERNLM-2 models, were evaluated for contamination | Apache 2.0 License |
| OLMO [25] | Includes all OLMO models, were evaluated for contamination | Apache 2.0 License |
| YI [53] | Includes all YI models, were evaluated for contamination | Yi Series Models Community License Agreement |
| GPT-4-TURBO | Used to generate synthetic and rephrased benchmarks | OpenAI Terms of Use |

Table 8: Contamination results for YAM-PELEG/EXPERIMENT26-7b, MTSAIR/-MULTI_VERSE_MODEL, BARRAHOME/MISTROLL-7b-v2.2. B is benchmark-specific, S is sample-specific and Y is syntax-specific contamination. All numbers are reported in percentages.

| Model | Benchmark | Perf. [%] | Type | $p$ [%] | $\hat{\delta}$ [%] | $\hat{\delta}_{0.95}$ [%] |
|---|---|---|---|---|---|---|
| BARRAHOME/MISTROLL-7b-v2.2 | ARC | 72.53 | B | $< 10^{-2}$ | 22.21 | 17.06 |
| | | | S | 55.90 | $-0.22$ | $-4.88$ |
| | | | Y | 88.15 | $-3.09$ | $-7.26$ |
| | GSM8k | 74.53 | B | $< 10^{-2}$ | 29.28 | 17.77 |
| | | | S | 0.34 | 7.34 | 3.05 |
| | | | Y | 34.00 | 0.79 | $-2.49$ |
| | Hellaswag | 88.60 | B | $< 10^{-2}$ | 12.95 | 10.15 |
| | | | S | 96.48 | $-3.25$ | $-6.01$ |
| | | | Y | 12.88 | 1.35 | $-0.57$ |
| | MMLU | 58.89 | S | 35.65 | 0.71 | $-2.41$ |
| | | | Y | 85.66 | $-1.83$ | $-4.62$ |
| MTSAIR/MULTI_VERSE_MODEL | ARC | 72.44 | B | $< 10^{-2}$ | 22.12 | 16.97 |
| | | | S | 51.07 | 0.13 | $-4.49$ |
| | | | Y | 87.29 | $-2.91$ | $-6.99$ |
| | GSM8k | 74.68 | B | $< 10^{-2}$ | 29.84 | 18.43 |
| | | | S | 0.53 | 7.20 | 2.65 |
| | | | Y | 47.26 | 0.12 | $-3.02$ |
| | Hellaswag | 88.55 | B | $< 10^{-2}$ | 12.14 | 9.56 |
| | | | S | 96.40 | $-3.19$ | $-5.92$ |
| | | | Y | 10.12 | 1.50 | $-0.40$ |
| | MMLU | 58.98 | S | 33.99 | 0.82 | $-2.33$ |
| | | | Y | 81.48 | $-1.52$ | $-4.30$ |
| YAM-PELEG/EXPERIMENT26-7b | ARC | 72.44 | B | $< 10^{-2}$ | 22.36 | 17.28 |
| | | | S | 60.10 | $-0.53$ | $-5.26$ |
| | | | Y | 90.97 | $-3.81$ | $-8.09$ |
| | GSM8k | 74.53 | B | $< 10^{-2}$ | 29.69 | 18.15 |
| | | | S | 0.24 | 7.60 | 3.19 |
| | | | Y | 14.74 | 2.07 | $-1.19$ |
| | Hellaswag | 88.60 | B | $< 10^{-2}$ | 13.11 | 10.27 |
| | | | S | 96.52 | $-3.25$ | $-6.00$ |
| | | | Y | 13.35 | 1.36 | $-0.59$ |
| | MMLU | 59.03 | S | 23.81 | 1.36 | $-1.79$ |
| | | | Y | 78.91 | $-1.36$ | $-4.17$ |

# E  All Test Results

We present all results for each performed test in this section. Tables Table 8-Table 23 contain these results, grouped by model family.

Table 9: Contamination results for QWEN-INSTRUCT-1.5-4b, QWEN-INSTRUCT-1.5-7b, QWEN-INSTRUCT-1.5-1.8b. B is benchmark-specific, S is sample-specific and Y is syntax-specific contamination. All numbers are reported in percentages.

| Model | Benchmark | Perf. [%] | Type | $p$ [%] | $\hat{\delta}$ [%] | $\hat{\delta}_{0.95}$ [%] |
|---|---|---|---|---|---|---|
| QWEN-INSTRUCT-1.5-1.8b | ARC | 38.40 | B | 97.39 | −6.47 | −12.37 |
| | | | S | 65.23 | −1.17 | −6.56 |
| | | | Y | 75.55 | −1.38 | −4.72 |
| | GSM8k | 31.16 | B | 16.59 | 6.43 | −4.08 |
| | | | S | 26.48 | 1.46 | −2.38 |
| | | | Y | 5.62 | 3.43 | −0.13 |
| | Hellaswag | 60.90 | B | 100.00 | −9.29 | −13.98 |
| | | | S | 99.73 | −7.34 | −11.27 |
| | | | Y | 82.43 | −2.45 | −7.91 |
| | MMLU | 41.69 | S | 34.60 | 0.56 | −2.43 |
| | | | Y | 32.23 | 0.79 | −2.15 |
| QWEN-INSTRUCT-1.5-4b | ARC | 42.06 | B | 30.33 | 6.03 | −22.11 |
| | | | S | 32.46 | 0.79 | −3.96 |
| | | | Y | 86.90 | −2.10 | −5.24 |
| | GSM8k | 6.52 | B | 99.82 | −25.43 | −36.93 |
| | | | S | $< 10^{-2}$ | 5.35 | 4.01 |
| | | | Y | 11.87 | 1.59 | −0.61 |
| | Hellaswag | 69.35 | B | 93.97 | −3.53 | −7.06 |
| | | | S | 66.77 | −0.85 | −4.22 |
| | | | Y | 26.14 | 1.36 | −3.00 |
| | MMLU | 50.36 | S | 45.01 | 0.25 | −2.87 |
| | | | Y | 88.31 | −2.14 | −5.09 |
| QWEN-INSTRUCT-1.5-7b | ARC | 55.12 | B | 3.05 | 14.29 | 2.24 |
| | | | S | 59.77 | −0.47 | −3.61 |
| | | | Y | 57.67 | −0.33 | −3.28 |
| | GSM8k | 55.12 | B | 59.20 | −1.38 | −11.91 |
| | | | S | 9.79 | 3.73 | −1.09 |
| | | | Y | 29.53 | 1.16 | −2.54 |
| | Hellaswag | 78.65 | B | 0.74 | 7.46 | 3.00 |
| | | | S | 90.61 | −1.87 | −4.18 |
| | | | Y | 15.20 | 1.74 | −1.16 |
| | MMLU | 57.82 | S | 69.80 | −1.02 | −4.37 |
| | | | Y | 50.88 | −0.02 | −3.04 |

Table 10: Contamination results for QWEN-INSTRUCT-1.5-14b, QWEN-INSTRUCT-1.5-72b, QWEN-INSTRUCT-1.5-110b. B is benchmark-specific, S is sample-specific and Y is syntax-specific contamination. All numbers are reported in percentages.

| Model | Benchmark | Perf. [%] | Type | $p$ [%] | $\hat{\delta}$ [%] | $\hat{\delta}_{0.95}$ [%] |
|---|---|---|---|---|---|---|
| QWEN-INSTRUCT-1.5-110b | ARC | 69.45 | B | 0.12 | 9.33 | 4.47 |
| | | | S | 54.00 | −0.20 | −4.13 |
| | | | Y | 56.38 | −0.25 | −3.29 |
| | GSM8k | 82.11 | B | 79.77 | −2.15 | −7.64 |
| | | | S | 20.01 | 2.05 | −1.94 |
| | | | Y | 7.08 | 2.80 | −0.35 |
| | Hellaswag | 84.25 | B | 29.03 | 0.69 | −1.48 |
| | | | S | 93.86 | −2.31 | −4.71 |
| | | | Y | 25.42 | 0.93 | −1.50 |
| | MMLU | 74.00 | S | 65.01 | −0.85 | −4.87 |
| | | | Y | 67.92 | −0.72 | −3.40 |
| QWEN-INSTRUCT-1.5-14b | ARC | 56.91 | B | 0.71 | 11.77 | 5.94 |
| | | | S | 88.65 | −2.50 | −5.72 |
| | | | Y | 32.93 | 0.77 | −2.11 |
| | GSM8k | 68.39 | B | 5.12 | 10.28 | −0.04 |
| | | | S | 1.13 | 6.37 | 1.89 |
| | | | Y | 75.33 | −1.45 | −4.94 |
| | Hellaswag | 82.15 | B | 0.07 | 6.48 | 3.74 |
| | | | S | 98.89 | −3.14 | −5.39 |
| | | | Y | 3.41 | 2.61 | 0.27 |
| | MMLU | 64.94 | S | 70.02 | −1.12 | −4.76 |
| | | | Y | 34.23 | 0.67 | −2.02 |
| QWEN-INSTRUCT-1.5-72b | ARC | 64.68 | B | 0.01 | 12.59 | 7.61 |
| | | | S | 70.96 | −1.34 | −5.42 |
| | | | Y | 6.07 | 2.77 | −0.17 |
| | GSM8k | 79.45 | B | 5.76 | 9.74 | −0.42 |
| | | | S | 38.33 | 0.77 | −3.41 |
| | | | Y | 98.90 | −4.39 | −7.54 |
| | Hellaswag | 86.35 | B | $< 10^{-2}$ | 8.24 | 6.05 |
| | | | S | 14.24 | 1.48 | −0.76 |
| | | | Y | 27.24 | 0.76 | −1.31 |
| | MMLU | 74.29 | S | 11.91 | 2.65 | −1.18 |
| | | | Y | 22.85 | 1.23 | −1.48 |

Table 11: Contamination results for OLMO-INSTRUCT-7b. B is benchmark-specific, S is sample-specific and Y is syntax-specific contamination. All numbers are reported in percentages.

| Model | Benchmark | Perf. [%] | Type | $p$ [%] | $\hat{\delta}$ [%] | $\hat{\delta}_{0.95}$ [%] |
|---|---|---|---|---|---|---|
| OLMO-INSTRUCT-7b | ARC | 46.08 | B | 22.68 | 7.22 | −11.30 |
| | | | S | 45.29 | 0.16 | −2.93 |
| | | | Y | 53.61 | −0.10 | −3.10 |
| | GSM8k | 11.75 | B | $< 10^{-2}$ | 8.86 | 4.99 |
| | | | S | 7.31 | 2.14 | −0.33 |
| | | | Y | 57.97 | −0.25 | −2.52 |
| | Hellaswag | 79.95 | B | 2.99 | 2.63 | 0.41 |
| | | | S | 5.04 | 2.38 | 0.00 |
| | | | Y | 17.90 | 1.14 | −1.08 |
| | MMLU | 41.94 | S | 70.97 | −0.69 | −2.88 |
| | | | Y | 48.80 | 0.03 | −2.61 |

Table 12: Contamination results for GEMMA-1.1-INSTRUCT-2b, GEMMA-1.1-INSTRUCT-7b. B is benchmark-specific, S is sample-specific and Y is syntax-specific contamination. All numbers are reported in percentages.

| Model | Benchmark | Perf. [%] | Type | $p$ [%] | $\hat{\delta}$ [%] | $\hat{\delta}_{0.95}$ [%] |
|---|---|---|---|---|---|---|
| GEMMA-1.1-INSTRUCT-2b | ARC | 44.71 | B | 99.47 | −7.12 | −11.79 |
| | | | S | 26.13 | 0.69 | −2.88 |
| | | | Y | 73.27 | −0.96 | −3.51 |
| | GSM8k | 10.61 | B | 100.00 | −28.17 | −38.87 |
| | | | S | 90.87 | −1.80 | −3.67 |
| | | | Y | 95.11 | −2.43 | −4.72 |
| | Hellaswag | 63.25 | B | 89.68 | −4.47 | −9.01 |
| | | | S | 87.14 | −2.47 | −5.07 |
| | | | Y | 21.24 | 0.64 | −4.85 |
| | MMLU | 35.64 | S | 74.14 | −0.93 | −3.74 |
| | | | Y | 85.61 | −1.75 | −4.17 |
| GEMMA-1.1-INSTRUCT-7b | ARC | 58.02 | B | 9.31 | 3.64 | −1.02 |
| | | | S | 10.48 | 2.13 | −0.74 |
| | | | Y | 44.06 | 0.21 | −2.44 |
| | GSM8k | 50.80 | B | 98.63 | −13.67 | −23.14 |
| | | | S | 81.05 | −2.23 | −6.10 |
| | | | Y | 64.59 | −0.72 | −4.00 |
| | Hellaswag | 76.85 | B | 1.79 | 5.79 | 1.59 |
| | | | S | 3.21 | 3.18 | 0.40 |
| | | | Y | 13.92 | 1.76 | −1.16 |
| | MMLU | 53.70 | S | 58.46 | −0.31 | −3.06 |
| | | | Y | 90.68 | −2.09 | −4.58 |

Table 13: Contamination results for INTERNLM-2-7b, INTERNLM-2-MATH-7b, INTERNLM-2-MATH-BASE-7b, INTERNLM-2-1.8b. B is benchmark-specific, S is sample-specific and Y is syntax-specific contamination. All numbers are reported in percentages.

| Model | Benchmark | Perf. [%] | Type | $p$ [%] | $\hat{\delta}$ [%] | $\hat{\delta}_{0.95}$ [%] |
|---|---|---|---|---|---|---|
| INTERNLM-2-1.8b | ARC | 39.93 | B | 100.00 | −19.05 | −23.95 |
| | | | S | 85.40 | −2.29 | −6.76 |
| | | | Y | 75.13 | −1.28 | −4.52 |
| | GSM8k | 24.03 | B | 5.93 | 7.98 | −0.37 |
| | | | S | 19.37 | 1.64 | −1.62 |
| | | | Y | 91.30 | −2.63 | −5.83 |
| | Hellaswag | 63.05 | B | 98.31 | −6.84 | −11.65 |
| | | | S | 43.03 | 0.03 | −5.80 |
| | | | Y | 40.35 | 0.33 | −5.29 |
| | MMLU | 41.26 | S | 82.17 | −1.54 | −4.35 |
| | | | Y | 52.23 | −0.13 | −3.12 |
| INTERNLM-2-7b | ARC | 55.55 | B | 100.00 | −13.68 | −18.74 |
| | | | S | 3.51 | 3.44 | 0.34 |
| | | | Y | 27.51 | 0.92 | −1.79 |
| | GSM8k | 62.09 | B | 0.43 | 19.27 | 7.98 |
| | | | S | 11.63 | 3.17 | −1.21 |
| | | | Y | 84.26 | −2.10 | −5.58 |
| | Hellaswag | 80.10 | B | 0.41 | 6.58 | 3.19 |
| | | | S | 2.64 | 2.73 | 0.46 |
| | | | Y | 30.02 | 0.65 | −1.53 |
| | MMLU | 57.77 | S | 33.66 | 0.80 | −2.38 |
| | | | Y | 9.90 | 2.20 | −0.62 |
| INTERNLM-2-MATH-7b | ARC | 52.82 | B | 84.86 | −3.14 | −8.28 |
| | | | S | 41.52 | 0.41 | −3.09 |
| | | | Y | 21.28 | 1.52 | −1.61 |
| | GSM8k | 72.93 | B | $< 10^{-2}$ | 39.40 | 27.15 |
| | | | S | 64.34 | −0.89 | −4.87 |
| | | | Y | 71.92 | −1.16 | −4.36 |
| | Hellaswag | 77.65 | B | 0.90 | 8.55 | 3.11 |
| | | | S | 5.57 | 2.80 | −0.11 |
| | | | Y | 33.90 | 0.64 | −2.48 |
| | MMLU | 57.48 | S | 33.57 | 0.86 | −2.49 |
| | | | Y | 53.43 | −0.14 | −3.15 |
| INTERNLM-2-MATH-BASE-7b | ARC | 56.23 | B | 64.31 | −0.99 | −5.50 |
| | | | S | 21.60 | 1.49 | −1.68 |
| | | | Y | 38.46 | 0.52 | −2.32 |
| | GSM8k | 35.63 | B | 83.69 | −7.17 | −18.75 |
| | | | S | 1.48 | 5.45 | 1.57 |
| | | | Y | 14.23 | 2.36 | −1.25 |
| | Hellaswag | 79.65 | B | 0.40 | 11.41 | 5.51 |
| | | | S | 4.63 | 2.41 | 0.05 |
| | | | Y | 35.40 | 0.48 | −1.80 |
| | MMLU | 52.83 | S | 48.63 | 0.08 | −3.16 |
| | | | Y | 11.60 | 2.36 | −0.93 |

Table 14: Contamination results for LLAMA-2-INSTRUCT-7b, LLAMA-2-7b, LLAMA-2-INSTRUCT-13b, LLAMA-2-13b, LLAMA-2-INSTRUCT-70b. B is benchmark-specific, S is sample-specific and Y is syntax-specific contamination. All numbers are reported in percentages.

| Model | Benchmark | Perf. [%] | Type | $p$ [%] | $\hat{\delta}$ [%] | $\hat{\delta}_{0.95}$ [%] |
|---|---|---|---|---|---|---|
| LLAMA-2-13b | ARC | 56.23 | B | 98.26 | $-5.46$ | $-9.43$ |
| | | | S | 13.56 | 1.62 | $-0.92$ |
| | | | Y | 48.39 | 0.07 | $-2.02$ |
| | GSM8k | 23.81 | B | 72.02 | $-3.41$ | $-13.58$ |
| | | | S | 13.14 | 1.74 | $-1.24$ |
| | | | Y | 78.51 | $-1.33$ | $-4.12$ |
| | Hellaswag | 82.35 | B | 97.33 | $-2.01$ | $-3.69$ |
| | | | S | 2.07 | 2.38 | 0.51 |
| | | | Y | 51.41 | $-0.03$ | $-1.49$ |
| | MMLU | 48.52 | S | 73.33 | $-0.88$ | $-3.21$ |
| | | | Y | 70.56 | $-0.76$ | $-3.14$ |
| LLAMA-2-7b | ARC | 52.56 | B | 59.44 | $-0.66$ | $-5.10$ |
| | | | S | 4.07 | 2.89 | 0.21 |
| | | | Y | 18.31 | 1.14 | $-1.06$ |
| | GSM8k | 13.42 | B | 38.75 | 1.60 | $-5.56$ |
| | | | S | 35.13 | 0.37 | $-1.68$ |
| | | | Y | 53.37 | $-0.09$ | $-2.38$ |
| | Hellaswag | 78.95 | B | 98.22 | $-2.42$ | $-4.35$ |
| | | | S | 5.41 | 2.03 | $-0.05$ |
| | | | Y | 44.09 | 0.13 | $-1.66$ |
| | MMLU | 41.31 | S | 7.68 | 2.91 | $-0.54$ |
| | | | Y | 14.99 | 1.54 | $-1.14$ |
| LLAMA-2-INSTRUCT-13b | ARC | 56.57 | B | 3.70 | 4.98 | 0.49 |
| | | | S | 72.99 | $-0.89$ | $-3.46$ |
| | | | Y | 78.67 | $-1.15$ | $-3.56$ |
| | GSM8k | 36.69 | B | 35.55 | 2.70 | $-8.22$ |
| | | | S | 35.35 | 0.62 | $-2.92$ |
| | | | Y | 35.94 | 0.66 | $-2.46$ |
| | Hellaswag | 82.35 | B | 77.82 | $-0.85$ | $-2.75$ |
| | | | S | 18.86 | 1.05 | $-0.95$ |
| | | | Y | 10.86 | 1.30 | $-0.47$ |
| | MMLU | 46.83 | S | 80.09 | $-1.26$ | $-3.55$ |
| | | | Y | 86.45 | $-1.72$ | $-4.08$ |
| LLAMA-2-INSTRUCT-70b | ARC | 61.86 | B | 28.49 | 1.34 | $-2.70$ |
| | | | S | 14.68 | 1.96 | $-1.01$ |
| | | | Y | 87.74 | $-1.77$ | $-4.21$ |
| | GSM8k | 55.80 | B | 7.23 | 9.18 | $-1.11$ |
| | | | S | 58.60 | $-0.41$ | $-4.42$ |
| | | | Y | 57.70 | $-0.33$ | $-3.55$ |
| | Hellaswag | 85.55 | B | 6.44 | 1.64 | $-0.15$ |
| | | | S | 0.41 | 3.37 | 1.29 |
| | | | Y | 71.66 | $-0.52$ | $-2.11$ |
| | MMLU | 56.85 | S | 32.91 | 0.71 | $-2.10$ |
| | | | Y | 43.26 | 0.25 | $-2.25$ |
| LLAMA-2-INSTRUCT-7b | ARC | 51.02 | B | 4.74 | 5.01 | 0.13 |
| | | | S | 82.83 | $-1.52$ | $-4.05$ |
| | | | Y | 76.26 | $-0.96$ | $-3.19$ |
| | GSM8k | 22.74 | B | 16.17 | 5.40 | $-2.92$ |
| | | | S | 68.05 | $-0.59$ | $-3.00$ |
| | | | Y | 7.90 | 2.35 | $-0.41$ |
| | Hellaswag | 78.10 | B | 91.52 | $-1.61$ | $-3.52$ |
| | | | S | 93.08 | $-1.88$ | $-3.89$ |
| | | | Y | 53.49 | $-0.12$ | $-2.13$ |
| | MMLU | 43.24 | S | 86.64 | $-1.67$ | $-3.78$ |
| | | | Y | 43.76 | 0.20 | $-2.22$ |

Table 15: Contamination results for LLAMA-3-INSTRUCT-70b, LLAMA-3-70b, LLAMA-3-8b, LLAMA-3-INSTRUCT-8b. B is benchmark-specific, S is sample-specific and Y is syntax-specific contamination. All numbers are reported in percentages.

| Model | Benchmark | Perf. [%] | Type | $p$ [%] | $\hat{\delta}$ [%] | $\hat{\delta}_{0.95}$ [%] |
|---|---|---|---|---|---|---|
| LLAMA-3-70b | ARC | 69.03 | B | 46.06 | 0.26 | −4.05 |
| | | | S | 0.03 | 6.61 | 3.21 |
| | | | Y | 1.95 | 3.35 | 0.73 |
| | GSM8k | 81.58 | B | 92.86 | −5.03 | −9.55 |
| | | | S | 15.45 | 2.05 | −1.49 |
| | | | Y | 23.24 | 1.13 | −1.60 |
| | Hellaswag | 86.45 | B | 77.74 | −0.74 | −2.43 |
| | | | S | 1.01 | 2.86 | 0.90 |
| | | | Y | 94.19 | −2.01 | −3.82 |
| | MMLU | 76.46 | S | 5.76 | 3.35 | −0.21 |
| | | | Y | 55.02 | −0.12 | −2.21 |
| LLAMA-3-8b | ARC | 56.91 | B | 99.41 | −6.46 | −10.30 |
| | | | S | 62.47 | −0.39 | −2.68 |
| | | | Y | 49.13 | 0.04 | −2.09 |
| | GSM8k | 49.36 | B | 99.38 | −15.27 | −24.75 |
| | | | S | 7.20 | 4.06 | −0.59 |
| | | | Y | 1.94 | 4.52 | 1.04 |
| | Hellaswag | 80.90 | B | 99.26 | −2.66 | −4.35 |
| | | | S | 44.87 | 0.15 | −1.70 |
| | | | Y | 94.78 | −1.48 | −2.94 |
| | MMLU | 61.16 | S | 4.18 | 3.01 | 0.18 |
| | | | Y | 10.88 | 1.78 | −0.63 |
| LLAMA-3-INSTRUCT-70b | ARC | 70.22 | B | 28.64 | 1.45 | −2.80 |
| | | | S | 75.48 | −1.37 | −5.12 |
| | | | Y | 68.86 | −0.52 | −2.79 |
| | GSM8k | 89.99 | B | 73.62 | −1.21 | −7.54 |
| | | | S | 9.39 | 3.05 | −1.03 |
| | | | Y | 20.00 | 1.23 | −1.50 |
| | Hellaswag | 85.30 | B | 18.97 | 0.87 | −0.88 |
| | | | S | 97.73 | −2.45 | −4.38 |
| | | | Y | 89.67 | −1.24 | −2.67 |
| | MMLU | 77.53 | S | 43.27 | 0.17 | −3.56 |
| | | | Y | 15.82 | 1.18 | −1.07 |
| LLAMA-3-INSTRUCT-8b | ARC | 60.75 | B | 45.04 | 0.25 | −3.77 |
| | | | S | 66.87 | −0.77 | −3.88 |
| | | | Y | 44.33 | 0.20 | −2.09 |
| | GSM8k | 75.89 | B | 7.93 | 8.40 | −1.20 |
| | | | S | 23.07 | 1.49 | −2.25 |
| | | | Y | 70.85 | −0.89 | −3.63 |
| | Hellaswag | 78.10 | B | 99.46 | −3.02 | −4.95 |
| | | | S | 98.96 | −2.96 | −4.85 |
| | | | Y | 90.89 | −1.43 | −3.18 |
| | MMLU | 62.23 | S | 76.69 | −1.17 | −3.80 |
| | | | Y | 27.51 | 0.74 | −1.53 |

Table 16: Contamination results for PHI-2, PHI-3-MINI, PHI-3-SMALL, PHI-3-MEDIUM. B is benchmark-specific, S is sample-specific and Y is syntax-specific contamination. All numbers are reported in percentages.

| Model | Benchmark | Perf. [%] | Type | $p$ [%] | $\hat{\delta}$ [%] | $\hat{\delta}_{0.95}$ [%] |
|---|---|---|---|---|---|---|
| PHI-2 | ARC | 58.45 | B | 70.14 | −1.29 | −5.43 |
| | | | S | 97.63 | −4.51 | −7.80 |
| | | | Y | 53.03 | −0.08 | −2.39 |
| | GSM8k | 58.91 | B | $< 10^{-2}$ | 36.42 | 26.46 |
| | | | S | 56.18 | −0.26 | −4.03 |
| | | | Y | 28.20 | 1.04 | −2.09 |
| | Hellaswag | 76.30 | B | 4.64 | 3.46 | 0.08 |
| | | | S | 100.00 | −4.80 | −6.73 |
| | | | Y | 26.31 | 0.83 | −1.68 |
| | MMLU | 51.96 | S | 99.39 | −5.01 | −7.87 |
| | | | Y | 40.50 | 0.32 | −2.16 |
| PHI-3-MINI | ARC | 59.90 | B | 59.70 | −0.61 | −4.72 |
| | | | S | 99.78 | −6.94 | −10.60 |
| | | | Y | 49.33 | 0.05 | −2.51 |
| | GSM8k | 76.65 | B | 0.29 | 16.30 | 6.33 |
| | | | S | 89.75 | −2.86 | −6.29 |
| | | | Y | 86.35 | −1.73 | −4.29 |
| | Hellaswag | 80.55 | B | 1.15 | 3.38 | 1.17 |
| | | | S | 100.00 | −6.34 | −8.37 |
| | | | Y | 62.19 | −0.35 | −2.26 |
| | MMLU | 63.49 | S | 99.01 | −4.94 | −8.05 |
| | | | Y | 40.40 | 0.29 | −2.15 |
| PHI-3-SMALL | ARC | 67.83 | B | 74.84 | −1.88 | −6.62 |
| | | | S | 63.09 | −0.76 | −4.55 |
| | | | Y | 26.75 | 1.15 | −1.84 |
| | GSM8k | 87.95 | B | 0.58 | 15.04 | 5.04 |
| | | | S | 61.30 | −0.70 | −5.04 |
| | | | Y | 58.49 | −0.36 | −3.31 |
| | Hellaswag | 84.70 | B | 0.02 | 5.41 | 3.38 |
| | | | S | 81.51 | −1.14 | −3.27 |
| | | | Y | 71.57 | −0.60 | −2.40 |
| | MMLU | 71.33 | S | 89.80 | −2.89 | −6.67 |
| | | | Y | 40.60 | 0.37 | −2.29 |
| PHI-3-MEDIUM | ARC | 66.55 | B | 83.99 | −2.73 | −7.16 |
| | | | S | 87.56 | −2.68 | −6.57 |
| | | | Y | 91.94 | −2.57 | −5.55 |
| | GSM8k | 86.35 | B | 0.03 | 21.11 | 10.78 |
| | | | S | 37.34 | 0.76 | −3.23 |
| | | | Y | 65.15 | −0.66 | −3.59 |
| | Hellaswag | 86.05 | B | 0.02 | 4.54 | 2.51 |
| | | | S | 92.44 | −1.83 | −3.87 |
| | | | Y | 20.51 | 0.87 | −0.89 |
| | MMLU | 74.14 | S | 63.19 | −0.73 | −4.64 |
| | | | Y | 51.35 | −0.04 | −2.64 |

Table 17: Contamination results for MISTRAL-7b-v0.1, MISTRAL-7b-v0.2. B is benchmark-specific, S is sample-specific and Y is syntax-specific contamination. All numbers are reported in percentages.

| Model | Benchmark | Perf. [%] | Type | $p$ [%] | $\hat{\delta}$ [%] | $\hat{\delta}_{0.95}$ [%] |
|---|---|---|---|---|---|---|
| MISTRAL-7b-v0.1 | ARC | 58.96 | B | 95.01 | −4.46 | −8.76 |
| | | | S | 7.91 | 2.21 | −0.40 |
| | | | Y | 28.07 | 0.79 | −1.52 |
| | GSM8k | 39.04 | B | 87.59 | −7.71 | −18.04 |
| | | | S | 0.15 | 8.25 | 4.48 |
| | | | Y | 66.44 | −0.87 | −4.42 |
| | Hellaswag | 83.65 | B | 65.34 | −0.42 | −2.19 |
| | | | S | 0.24 | 3.14 | 1.27 |
| | | | Y | 16.97 | 0.94 | −0.68 |
| | MMLU | 58.01 | S | 15.23 | 1.88 | −1.05 |
| | | | Y | 11.08 | 1.88 | −0.69 |
| MISTRAL-7b-v0.2 | ARC | 58.19 | B | 98.02 | −5.63 | −10.01 |
| | | | S | 23.24 | 1.17 | −1.38 |
| | | | Y | 45.61 | 0.15 | −2.11 |
| | GSM8k | 37.60 | B | 89.88 | −8.55 | −18.87 |
| | | | S | 0.54 | 6.57 | 2.68 |
| | | | Y | 76.32 | −1.53 | −5.07 |
| | Hellaswag | 82.80 | B | 58.14 | −0.25 | −2.11 |
| | | | S | 1.28 | 2.60 | 0.69 |
| | | | Y | 50.08 | −0.01 | −1.60 |
| | MMLU | 56.71 | S | 49.62 | 0.06 | −2.87 |
| | | | Y | 27.94 | 0.89 | −1.61 |

Table 18: Contamination results for MISTRAL-INSTRUCT-7b-v0.3, MISTRAL-INSTRUCT-7b-v0.2, MISTRAL-INSTRUCT-7b-v0.1. B is benchmark-specific, S is sample-specific and Y is syntax-specific contamination. All numbers are reported in percentages.

| Model | Benchmark | Perf. [%] | Type | $p$ [%] | $\hat{\delta}$ [%] | $\hat{\delta}_{0.95}$ [%] |
|---|---|---|---|---|---|---|
| MISTRAL-INSTRUCT-7b-v0.1 | ARC | 54.35 | B | 99.97 | −9.04 | −12.92 |
| | | | S | 95.10 | −2.83 | −5.40 |
| | | | Y | 73.29 | −0.86 | −3.17 |
| | GSM8k | 35.86 | B | 98.51 | −13.89 | −23.86 |
| | | | S | 85.29 | −2.45 | −5.63 |
| | | | Y | 90.91 | −2.67 | −5.83 |
| | Hellaswag | 74.90 | B | 90.11 | −1.68 | −3.97 |
| | | | S | 97.12 | −2.47 | −4.62 |
| | | | Y | 53.04 | −0.13 | −2.51 |
| | MMLU | 49.69 | S | 29.29 | 0.77 | −1.76 |
| | | | Y | 72.16 | −0.87 | −3.36 |
| MISTRAL-INSTRUCT-7b-v0.2 | ARC | 62.46 | B | 0.04 | 10.62 | 5.95 |
| | | | S | 64.89 | −0.70 | −4.17 |
| | | | Y | 80.34 | −1.29 | −3.80 |
| | GSM8k | 42.61 | B | 80.76 | −5.71 | −16.01 |
| | | | S | 94.39 | −4.26 | −8.35 |
| | | | Y | 84.91 | −2.10 | −5.24 |
| | Hellaswag | 84.55 | B | 0.18 | 3.52 | 1.56 |
| | | | S | 81.26 | −1.04 | −3.01 |
| | | | Y | 12.47 | 1.10 | −0.56 |
| | MMLU | 55.06 | S | 18.63 | 1.48 | −1.28 |
| | | | Y | 42.67 | 0.25 | −2.21 |
| MISTRAL-INSTRUCT-7b-v0.3 | ARC | 62.37 | B | 88.82 | −3.24 | −7.59 |
| | | | S | 94.57 | −3.64 | −7.38 |
| | | | Y | 66.69 | −0.62 | −3.06 |
| | GSM8k | 48.22 | B | 94.33 | −10.60 | −20.56 |
| | | | S | 35.14 | 1.03 | −3.81 |
| | | | Y | 89.74 | −2.69 | −6.15 |
| | Hellaswag | 84.25 | B | 5.20 | 1.90 | −0.02 |
| | | | S | 62.58 | −0.40 | −2.42 |
| | | | Y | 37.49 | 0.28 | −1.31 |
| | MMLU | 56.56 | S | 14.15 | 1.88 | −0.93 |
| | | | Y | 34.01 | 0.63 | −1.89 |

Table 19: Contamination results for MIXTRAL-INSTRUCT-8x22b, MIXTRAL-INSTRUCT-8x7b. B is benchmark-specific, S is sample-specific and Y is syntax-specific contamination. All numbers are reported in percentages.

| Model | Benchmark | Perf. [%] | Type | $p$ [%] | $\hat{\delta}$ [%] | $\hat{\delta}_{0.95}$ [%] |
|---|---|---|---|---|---|---|
| MIXTRAL-INSTRUCT-8x22b | ARC | 72.95 | B | 0.77 | 6.01 | 2.00 |
| | | | S | 10.63 | 2.64 | −1.07 |
| | | | Y | 20.07 | 1.10 | −1.42 |
| | GSM8k | 85.97 | B | 44.60 | 0.69 | −3.86 |
| | | | S | 65.36 | −0.68 | −4.37 |
| | | | Y | 21.06 | 1.18 | −1.46 |
| | Hellaswag | 89.00 | B | 14.01 | 1.19 | −0.62 |
| | | | S | 22.49 | 0.82 | −1.13 |
| | | | Y | 26.36 | 0.62 | −1.25 |
| | MMLU | 74.14 | S | 29.85 | 0.97 | −2.57 |
| | | | Y | 59.95 | −0.23 | −2.42 |
| MIXTRAL-INSTRUCT-8x7b | ARC | 69.54 | B | 73.09 | −1.49 | −5.59 |
| | | | S | 47.27 | 0.11 | −3.48 |
| | | | Y | 68.30 | −0.55 | −2.85 |
| | GSM8k | 65.66 | B | 95.56 | −8.84 | −18.06 |
| | | | S | 69.87 | −1.06 | −4.53 |
| | | | Y | 83.79 | −1.78 | −4.63 |
| | Hellaswag | 87.60 | B | 20.79 | 0.82 | −0.85 |
| | | | S | 14.57 | 1.15 | −0.76 |
| | | | Y | 6.31 | 1.34 | −0.10 |
| | MMLU | 66.73 | S | 26.75 | 1.07 | −1.93 |
| | | | Y | 77.86 | −0.98 | −3.09 |

Table 20: Contamination results for STABLELM-2-12b, STABLELM-2-ZEPHYR-3b, STABLELM-2-1.6b, STABLELM-2-ALPHA-7b-v2. B is benchmark-specific, S is sample-specific and Y is syntax-specific contamination. All numbers are reported in percentages.

| Model | Benchmark | Perf. [%] | Type | $p$ [%] | $\hat{\delta}$ [%] | $\hat{\delta}_{0.95}$ [%] |
|---|---|---|---|---|---|---|
| STABLELM-2-ALPHA-7b-v2 | ARC | 44.28 | B | 100.00 | −10.65 | −15.06 |
| | | | S | 60.46 | −0.58 | −3.99 |
| | | | Y | 45.14 | 0.14 | −2.76 |
| | GSM8k | 5.31 | B | 61.89 | −1.10 | −7.51 |
| | | | S | 27.98 | 0.65 | −1.12 |
| | | | Y | 90.92 | −1.72 | −3.80 |
| | Hellaswag | 77.20 | B | 100.00 | −5.41 | −7.36 |
| | | | S | 3.88 | 2.65 | 0.23 |
| | | | Y | 43.34 | 0.15 | −2.18 |
| | MMLU | 42.13 | S | 3.20 | 4.00 | 0.49 |
| | | | Y | 26.45 | 1.11 | −1.87 |
| STABLELM-2-ZEPHYR-3b | ARC | 44.80 | B | 23.74 | 5.20 | −10.17 |
| | | | S | 75.44 | −1.43 | −4.93 |
| | | | Y | 56.38 | −0.32 | −3.55 |
| | GSM8k | 51.63 | B | $< 10^{-2}$ | 48.78 | 44.34 |
| | | | S | 3.29 | 5.47 | 0.54 |
| | | | Y | 75.39 | −1.45 | −4.92 |
| | Hellaswag | 72.95 | B | 35.74 | 0.74 | −3.18 |
| | | | S | 100.00 | −7.45 | −9.83 |
| | | | Y | 44.96 | 0.15 | −3.67 |
| | MMLU | 40.29 | S | 83.87 | −1.61 | −4.45 |
| | | | Y | 35.87 | 0.63 | −2.46 |
| STABLELM-2-1.6b | ARC | 43.43 | B | 100.00 | −26.35 | −31.68 |
| | | | S | 14.90 | 2.41 | −2.41 |
| | | | Y | 56.10 | −0.30 | −3.49 |
| | GSM8k | 18.88 | B | $< 10^{-2}$ | 16.56 | 12.95 |
| | | | S | 23.94 | 1.22 | −1.64 |
| | | | Y | 49.96 | 0.00 | −2.92 |
| | Hellaswag | 71.05 | B | 100.00 | −5.16 | −7.61 |
| | | | S | 5.19 | 4.31 | −0.06 |
| | | | Y | 31.06 | 1.02 | −2.93 |
| | MMLU | 35.69 | S | 29.73 | 1.25 | −2.34 |
| | | | Y | 32.27 | 0.84 | −2.20 |
| STABLELM-2-12b | ARC | 59.47 | B | 99.97 | −14.01 | −19.28 |
| | | | S | 0.68 | 4.61 | 1.59 |
| | | | Y | 1.10 | 3.78 | 1.12 |
| | GSM8k | 58.00 | B | 0.39 | 17.92 | 6.96 |
| | | | S | 11.92 | 3.10 | −1.34 |
| | | | Y | 17.98 | 1.85 | −1.51 |
| | Hellaswag | 84.45 | B | 18.16 | 0.98 | −0.87 |
| | | | S | 2.96 | 2.16 | 0.25 |
| | | | Y | 30.69 | 0.49 | −1.15 |
| | MMLU | 56.51 | S | 77.34 | −1.46 | −4.67 |
| | | | Y | 72.72 | −0.98 | −3.70 |

Table 21: Contamination results for STABLELM-2-INSTRUCT-12b, STABLELM-2-INSTRUCT-1.6b. B is benchmark-specific, S is sample-specific and Y is syntax-specific contamination. All numbers are reported in percentages.

| Model | Benchmark | Perf. [%] | Type | $p$ [%] | $\hat{\delta}$ [%] | $\hat{\delta}_{0.95}$ [%] |
|---|---|---|---|---|---|---|
| STABLELM-2-INSTRUCT-1.6b | ARC | 42.66 | B | 29.38 | 5.53 | −19.03 |
| | | | S | 70.33 | −1.20 | −5.11 |
| | | | Y | 48.97 | 0.00 | −3.39 |
| | GSM8k | 42.00 | B | $< 10^{-2}$ | 27.79 | 19.27 |
| | | | S | 26.27 | 1.77 | −2.67 |
| | | | Y | 62.76 | −0.67 | −4.32 |
| | Hellaswag | 69.60 | B | 31.25 | 1.08 | −4.38 |
| | | | S | 100.00 | −5.99 | −8.50 |
| | | | Y | 68.68 | −1.21 | −5.21 |
| | MMLU | 38.31 | S | 61.21 | −0.66 | −4.05 |
| | | | Y | 9.37 | 2.50 | −0.64 |
| STABLELM-2-INSTRUCT-12b | ARC | 64.25 | B | 2.17 | 26.11 | 5.20 |
| | | | S | 67.88 | −1.15 | −5.28 |
| | | | Y | 18.72 | 1.61 | −1.31 |
| | GSM8k | 68.84 | B | $< 10^{-2}$ | 32.93 | 21.09 |
| | | | S | 17.74 | 2.37 | −1.86 |
| | | | Y | 32.26 | 0.88 | −2.41 |
| | Hellaswag | 86.25 | B | $< 10^{-2}$ | 7.04 | 4.88 |
| | | | S | 98.40 | −3.43 | −5.84 |
| | | | Y | 3.32 | 2.18 | 0.24 |
| | MMLU | 55.50 | S | 90.13 | −2.49 | −5.67 |
| | | | Y | 44.42 | 0.25 | −2.74 |

Table 22: Contamination results for FALCON-INSTRUCT-7b, FALCON-7b. B is benchmark-specific, S is sample-specific and Y is syntax-specific contamination. All numbers are reported in percentages.

| Model | Benchmark | Perf. [%] | Type | $p$ [%] | $\hat{\delta}$ [%] | $\hat{\delta}_{0.95}$ [%] |
|---|---|---|---|---|---|---|
| FALCON-7b | ARC | 46.84 | B | 99.96 | −8.77 | −12.77 |
| | | | S | 78.77 | −1.23 | −3.80 |
| | | | Y | 39.48 | 0.30 | −2.10 |
| | GSM8k | 4.25 | B | 95.19 | −8.40 | −15.69 |
| | | | S | 89.76 | −1.36 | −2.70 |
| | | | Y | 79.29 | −0.83 | −2.50 |
| | Hellaswag | 78.50 | B | 100.00 | −4.13 | −5.97 |
| | | | S | 2.47 | 2.67 | 0.44 |
| | | | Y | 62.41 | −0.32 | −2.09 |
| | MMLU | 26.10 | S | 68.27 | −0.39 | −2.13 |
| | | | Y | 16.74 | 1.35 | −1.48 |
| FALCON-INSTRUCT-7b | ARC | 45.05 | B | 71.90 | −1.48 | −5.71 |
| | | | S | 94.92 | −3.15 | −5.90 |
| | | | Y | 77.85 | −1.17 | −3.64 |
| | GSM8k | 4.40 | B | 94.97 | −8.50 | −15.86 |
| | | | S | 49.71 | 0.04 | −1.47 |
| | | | Y | 41.32 | 0.23 | −1.43 |
| | Hellaswag | 69.15 | B | 99.80 | −4.81 | −7.94 |
| | | | S | 84.96 | −1.67 | −4.17 |
| | | | Y | 95.80 | −3.43 | −6.23 |
| | MMLU | 26.30 | S | 31.59 | 0.38 | −1.52 |
| | | | Y | 54.78 | −0.16 | −2.63 |

Table 23: Contamination results for YI-34b, YI-6b. B is benchmark-specific, S is sample-specific and Y is syntax-specific contamination. All numbers are reported in percentages.

| Model | Benchmark | Perf. [%] | Type | $p$ [%] | $\hat{\delta}$ [%] | $\hat{\delta}_{0.95}$ [%] |
|---|---|---|---|---|---|---|
| YI-34b | ARC | 63.99 | B | 99.86 | −14.18 | −20.29 |
| | | | S | 0.20 | 5.00 | 2.12 |
| | | | Y | 16.29 | 1.58 | −1.06 |
| | GSM8k | 66.79 | B | 95.95 | −9.04 | −18.47 |
| | | | S | 6.10 | 4.29 | −0.30 |
| | | | Y | 59.86 | −0.52 | −3.85 |
| | Hellaswag | 86.15 | B | 0.14 | 3.96 | 1.89 |
| | | | S | $< 10^{-2}$ | 6.51 | 4.40 |
| | | | Y | 4.97 | 1.75 | 0.01 |
| | MMLU | 71.48 | S | 41.94 | 0.49 | −3.22 |
| | | | Y | 24.43 | 1.09 | −1.57 |
| YI-6b | ARC | 54.44 | B | 100.00 | −15.81 | −20.49 |
| | | | S | 4.82 | 3.03 | 0.03 |
| | | | Y | 40.58 | 0.35 | −2.22 |
| | GSM8k | 33.59 | B | 70.57 | −3.51 | −14.79 |
| | | | S | 28.13 | 1.35 | −2.50 |
| | | | Y | 22.92 | 1.53 | −1.91 |
| | Hellaswag | 77.30 | B | 51.87 | −0.10 | −2.27 |
| | | | S | 8.65 | 1.94 | −0.39 |
| | | | Y | 68.03 | −0.65 | −2.92 |
| | MMLU | 57.38 | S | 42.25 | 0.40 | −3.03 |
| | | | Y | 57.89 | −0.33 | −3.22 |

