# OpenReview forum: "ConStat: Performance-Based Contamination Detection in Large Language Models"
_NeurIPS.cc/2024/Conference — NeurIPS 2024 poster_

### Official Review · Reviewer_UkYs · 2024-07-02

**Soundness:** 2
**Presentation:** 2
**Contribution:** 2
**Rating:** 6
**Confidence:** 3

**Summary:**

Authors propose a new definition of contamination: "artificially inflated and non-generalizing benchmark performance" rather than "the inclusion of benchmark samples in the training data". They develop ConStat, a statistical method that reliably detects and quantifies contamination by comparing performance between a primary and reference benchmark relative to a set of reference models.  They demonstrate the effectiveness of ConStat through an evaluation of diverse model architectures, benchmarks, and contamination scenarios, and find high levels of contamination in multiple popular models.

**Strengths:**

•	A new method to detect benchmark contamination from a new perspective

•	Great performance on syntax and sample-specific contaminated models

•	Good motivation, presentation and organisation

**Weaknesses:**

•	The paper uses reference datasets that are synthetically composed, authors briefly comment on this in the limitations. More ellaboration on this is need it since it is one of the basis of the experiments. Authors could potentially estimate by human analysis what is the expected error on their approach.

•	Benchmark-specific contamination is complex and it is not covered with enough in-depth.

**Questions:**

•	There is a short paragraph explaining results on benchmark-specific contamination, could authors elaborate more? How does it compare to other methods?

•	Some statements need clarification, e.g.
“MathQA is a multiple-choice benchmark that requires the model  to answer immediately and therefore gives no room for this reasoning ability to shine.” ->•	There is a section on exposing the limitations. Maybe authors could further comment on the limitation of experiments due to computational complexity. what does it mean that requires the model to answer immediately? Could authors further support “no room for this reasoning ability to shine”

**Limitations:**

•	There is a section on exposing the limitations. Maybe authors could further comment on the limitation of experiments due to computational complexity.

---

> ### Author Rebuttal · Authors · 2024-08-06
>
> We thank the reviewer for their detailed review and insightful questions. We are happy to hear that they found our motivation, presentation, and organization very good, and the performance of our method great. Below, we address their questions.
>
> **Could you elaborate more on the effect of the synthetic nature of the synthetic data on your approach?**
> We first note that the low quality of the synthetic datasets would not significantly affect the predictions made by ConStat. For instance, the inclusion of a sample with an incorrect label would lead to incorrect answers by all models. The hardness correction function would adjust for this incorrectness automatically, without affecting the contamination prediction.
>
> Furthermore, we performed extensive manual checks to ensure high data quality and fidelity of the synthetic data. Specifically, we followed this procedure:
> - We first manually tested around 10 samples for each benchmark with several system prompts and checked the output quality of the model. We adjusted the prompts until we were satisfied with the quality of these 10 samples.
> - We then conducted a manual check of around 100 samples for each benchmark to identify common mistakes and ensure overall data quality. This process helped us detect and exclude some synthetic samples in the GSM8k dataset that did not have integer answers.
> - We performed deduplication by computing the samples with the highest 1-gram overlap for each synthetic sample. We manually checked around 50 samples and their 1-gram overlaps to set the threshold for excluding a sample. We used a similar approach to check whether a synthetic sample was included in the original test set.
>
> **Could you elaborate more on benchmark-specific contamination? How does it compare to other methods?**
> Benchmark-specific contamination is the most generic and least problematic type of contamination. As we explain in Section 2.2, it occurs when a model fails to generalize to benchmarks that measure performance on the same task as the original benchmark. This implies that a model is only effective on specific questions or formats in the original benchmarks. Users should be cautious when applying this model to real-world scenarios, as its performance may not be consistent across the entire task.
>
> However, we note that this inconsistency can arise for reasons other than information-flow-based contamination. For instance, a focus during training on the particular type of questions in the original benchmark can cause benchmark-specific contamination. We acknowledge this in the paper and are careful in drawing major conclusions from this type of contamination. We will clarify this point in the paper to ensure that readers understand the necessary caution in interpreting results regarding benchmark-specific contamination and the conclusions that can be drawn from it.
>
> **Could you clarify the statement “MathQA is a multiple-choice benchmark that requires the model to answer immediately and therefore gives no room for this reasoning ability to shine.”?**
> This statement refers to our use of the LM Evaluation Harness, a popular framework for language model evaluation. The LM Evaluation Harness implements MathQA as a multiple-choice benchmark, where the framework measures the perplexity of the answer options and selects the option with the lowest perplexity. In contrast, for GSM8k, the framework extracts the answer from a free-form response created by the language model. We believe the Phi model family performs better in free-form evaluations than in multiple-choice evaluations, leading to the conclusion in the statement. This distinction is unrelated to computational complexity since we would have evaluated MathQA using free-form answers if the LM Evaluation Harness implemented the benchmark like this.
>
> **Could you comment on the computational complexity of your method, e.g., in the context of the prior statement?**
> As explained above, the computational complexity is not related to the multiple-choice evaluation of MathQA. However, ConStat does require evaluating several models on multiple benchmarks, which entails some computational complexity. Despite this, the total cost of our entire evaluation, amounting to a couple of thousand dollars, is negligible compared to the cost of training a model and serving it via an inference API. Therefore, we believe this does not pose a serious limitation, as our method can be applied by any organization with reasonable funds.
>
> We hope our answers address all of the reviewer’s concerns, remain happy to answer any further questions, and look forward to the reviewer’s response.

---

### Official Review · Reviewer_hKkW · 2024-07-10

**Soundness:** 2
**Presentation:** 4
**Contribution:** 2
**Rating:** 5
**Confidence:** 4

**Summary:**

The paper proposes a performance-based approach to detect data contamination in large language models. First, a set of reference models is evaluated on the original benchmark and a proxy of it. Next, a difficulty correction function is fitted to find the relationship between the performance from the original benchmark and the proxy benchmark. Then, the performance of a target model is evaluated on the proxy benchmark, and using the correction function, the expected performance on the original benchmark is predicted. Lastly, the difference between the actual performance on the original benchmark and the expected performance is computed and checked for significance to label the target model as contaminated.

**Strengths:**

1. The paper is well-written.
2. The experiments are comprehensive.
3. The proposed method can be used with both open-weight and closed-weight models, which is important.

**Weaknesses:**

1. The main weakness of the proposed method is that it identifies data contamination based on changes in performance. Performance can change for several reasons, so it cannot be solely the indicator of contamination. In fact, the method assumes that any performance increase means contamination, leading to a high rate of false positives in situations where performance improvements come from other sources. For example, this method incorrectly detects contamination when performance improves due to unsupervised domain adaptation while it is not.

2. This method can only be applied to models that are similar to the reference models, especially in terms of size, architecture, and pre-training data, as it detects contamination with respect to these reference models. This limits its generality and makes it unsuitable for models that differ from the reference set. For example, if a target model just generalizes better than the reference models, the method incorrectly identifies this as contamination. Conversely, a contaminated model can be deemed uncontaminated if it cannot translate the contamination into improved performance, e.g., when a model is contaminated with a dataset but cannot follow instructions very well. Therefore, the proposed method measures the lack of generalization relative to the reference models rather than actual contamination. In short, this method does not guarantee whether a model has seen the datasets or not.

3. Building on the previous comment, this method does not actually measure contamination; it captures situations where excessive memorization has replaced generalization. In fact, the scenarios discussed in Section 2.2 **are not types of contamination**. Instead, they are examples of excessive memorization, which the method captures.

4. The results of the proposed method are relative, not absolute. In this method, detection is only meaningful when compared to the reference models. So, if the reference models are contaminated, the target model will still appear uncontaminated.

**Questions:**

**Question:**
1. In Table 1, what is the difference between "Shi et al. [40]" and "Shi [39]"?

**Comment:**
1. Lines 11-14 and 343-345: Your method does not quantify/measure contamination.

**Limitations:**

1. The outcomes cannot be interpreted individually. Specifically, detection heavily depends on the reference models and how they generalize/behave. This means that better generalization or steerability in models can be mistaken for contamination. Also, if the reference models are contaminated, this contamination can spread to the target models and go undetected.
2. The proposed method does not detect contamination, as none of the scenarios studied in Section 2.2 involve contamination. Instead, the method captures the lack of generalization or situations with excessive memorization.

---

> ### Author Rebuttal · Authors · 2024-08-06
>
> We thank the reviewer for their detailed review and insightful questions. We are pleased to hear that they found our paper well-written, our experiments comprehensive, and the applicability of our method to closed-source models interesting. Below, we address their questions.
>
> **Should contamination be viewed from a perspective of performance generalization?**
> Yes, we argue that information-flow-based contamination can be viewed from this perspective for two main reasons.
>
> First, in benchmark evaluation, the only relevant consequence of information-flow-based contamination is an artificial increase in performance on benchmarks. Although there are other areas of influence, such as detecting the use of copyrighted data, our focus lies specifically on its effect on performance measurements. Given the large datasets involved in training LLMs, some level of information-flow-based contamination is almost inevitable. Therefore, any model can be said to show some level of information-flow-based contamination. Thus, a detailed contamination report that includes its effect on benchmark performance is more interesting and easily obtainable using ConStat.
>
> Second, syntax- and sample-specific contamination imply information-flow-based contamination. As we argue in Q1 of our main reply, a model showing syntax- or sample-specific contamination can distinguish between two benchmarks drawn from the same or a very similar distribution. This is only possible if one of these benchmarks was seen during training. Otherwise, the model should treat the two benchmarks identically, resulting in similar performance on both.
>
> **Is it possible that performance changes for reasons other than information-flow-based contamination?**
> Please see our detailed answer in the main response.
>
> **Can the method only be applied to models that are similar to the reference models?**
> No, **none** of our experiments rely on this assumption. Our reference models encompass a range of architectures, sizes, and pre-training data. Therefore, our results demonstrating the accurate detection and estimation of performance-based contamination in Section 4.2 do not rely on similarity assumptions. This robustness also allows us to detect contamination in various model architectures and sizes in Section 4.3 and reproduce contamination results from some model providers without access to their training data.
>
> **What kind of contamination does the method measure? Does it measure excessive memorization rather than contamination?**
> Yes, our method redefines contamination based on its influence on performance, which can also be seen as excessive memorization. As argued above and in the paper, this is a key effect of information-flow-based contamination, and the only aspect worth measuring for the purpose of benchmark integrity. While ConStat might miss information-flow-based contamination that does not affect performance, this is by design, as such contamination has no practical consequences on benchmark integrity.
>
> **Would the method incorrectly detect contamination when the target model’s performance generalizes worse than the reference models due to reasons other than contamination?**
> For uncontaminated models, this should only occur when detecting benchmark-specific contamination. As previously argued, syntax- and sample-specific contamination imply information-flow-based contamination. Therefore, the absence of information-flow-based contamination also implies the absence of syntax- or sample-specific contamination. We acknowledge other factors influencing benchmark-specific contamination in the paper and are cautious in our conclusions about this type. However, measuring this contamination is important. If a model excels on specific benchmarks but performs poorly on others for the same task, it indicates problematic behavior. While this might not be information-flow based contamination, it may result from a focus on a subset of the task during training because of a drive to perform well on a specific benchmark. Benchmark-specific contamination can reveal this issue and indicate how representative reported scores are with regard to overall task performance.
>
> **Is measuring contamination relative to a set of reference models sufficient for contamination detection?**
> Please see our detailed answer in the main response.
>
> **In Table 1, what is the difference between "Shi et al. [40]" and "Shi [39]"?**
> These are two different methods by the same authors. Shi et al. [40], known as TopKMin, measure the perplexity of each token in the answer, retain the k% tokens with the highest perplexities, and average these to obtain a contamination measure. In contrast, Shi [39] generates 30 alternative completions of the first half of each question in the benchmark using an uncontaminated base model. It then measures the perplexities of these completions with the target model, and contamination is detected by the percentile of the actual answer's perplexity among these completions. If this percentile is consistently low, the model is deemed contaminated.
>
> We hope our answers address all of the reviewer’s concerns, remain happy to answer any further questions, and look forward to the reviewer’s response.

---

> > ### Comment · Reviewer_hKkW · 2024-08-12
> >
> > Thanks to the reviewer for their rebuttals. While the rebuttal somewhat addressed some of my concerns, I still believe the method’s applicability is limited due to its reliance on a set of reference models. Therefore, I will maintain my score.

---

> > > ### Author Response · Authors · 2024-08-12
> > >
> > > We thank the reviewer for engaging in the discussion and are happy to hear that we could address some of their points. Could the reviewer explain in what scenarios they believe ConStat’s applicability to be limited by the required set of reference models? In particular, we have demonstrated that a lack of generalization will not lead to falsely detected contamination, as mentioned in the reviewer's response. Furthermore, we believe that the reproduction of the contamination numbers of Llama-2-70b from the original paper without access to the training data  demonstrates that this limitation is minimal, if not negligible.

---

### Official Review · Reviewer_tnxc · 2024-07-11

**Soundness:** 3
**Presentation:** 4
**Contribution:** 4
**Rating:** 8
**Confidence:** 3

**Summary:**

The authors introduce a definition for “contamination” based on its outcome rather than its cause, unlike many previous approaches. The authors propose ConStat, a novel method for quantifying the contamination of a model on some benchmark and demonstrate that it outperforms other methodologies for detecting contamination.

**Strengths:**

- This paper was a pleasure to read. It is both written and structured in a clear way that communicates the ideas well
- The authors used ConStat to show strong contamination of some models from the Open LLM Leaderboard on specific benchmarks. This research can be used to build trust in our evaluations

**Weaknesses:**

The authors generated a synthetic version of each benchmark they used in their experiments. They described their methodology for doing so in Appendix C. I don’t believe the authors used a sufficiently rigorous method to ensure this generated dataset was sufficiently high-quality. For example, it is possible that some of their samples did not have a high 1-gram overlap but did ask similar questions. I believe a manual review of a subset of the dataset for duplication, and perhaps other traits, would be necessary to guarantee data quality. Though I view this as a minor weakness given that their methodology was successful despite potential problems with this dataset.

**Questions:**

1. Why did the authors not select a reference benchmark for MMLU?
2. Do the authors believe this methodology will work well when the measured task between the benchmark and reference benchmark is similar (e.g. coding) but the "form" between the benchmark and reference benchmark is significantly different, for example, multiple choice question answering evaluation vs. open-ended agentic evaluation?
3. What further research are the authors excited to see in this area? Are the authors planning on discussing this within the paper?

**Limitations:**

The authors sufficiently address this.

---

> ### Author Rebuttal · Authors · 2024-08-06
>
> We thank the reviewer for their detailed review and insightful questions. We are happy to hear that they found our paper a pleasure to read and that our research can be used to build trust in our evaluations. Below, we address their questions.
>
> **How did you ensure that the generated dataset had a sufficiently high quality? Is the deduplication process using 1-grams sufficient?**
> To ensure high data quality for rephrasing and synthetic sample generation, we performed the following procedure:
> - We manually tested around 10 samples for each benchmark with various system prompts, iteratively refining the prompts until we were satisfied with the output quality.
> - We performed a manual check of approximately 100 samples for each benchmark to identify common mistakes and evaluate overall data quality. This process revealed issues such as non-integer answers in some GSM8k samples, which were therefore excluded.
> - For deduplication, we computed the highest 1-gram overlaps between synthetic samples and manually reviewed around 50 samples to set an appropriate exclusion threshold. We used a similar method to ensure no synthetic samples were present in the original test set.
>
> It is important to note that low-quality synthetic data would not significantly impact ConStat’s predictions. For instance, a sample with an incorrect label would be consistently answered incorrectly by all models. The hardness correction function would adjust for this automatically, ensuring contamination prediction remains unaffected.
>
> Furthermore, the inclusion of test set samples in the synthetic data set would also only reduce the effectiveness of ConStat: these samples would reduce the estimated performance difference and therefore make ConStat’s estimates less accurate. The fact that our estimates are quite accurate, suggests that our approach was sufficient to prevent inclusion of test set samples in our synthetic data.
>
> **Why did the authors not select a reference benchmark for MMLU to measure benchmark-specific contamination?**
> MMLU is a broad benchmark designed to evaluate knowledge across diverse tasks. Benchmark-specific contamination detection requires a reference benchmark closely aligned with the target benchmark’s task. We could not find a sufficiently similar benchmark to MMLU, and including a dissimilar benchmark might lead to false positives. Therefore, we chose not to include benchmark-specific contamination detection for MMLU to maintain the accuracy of our analysis.
>
> **Do you expect your approach for measuring benchmark-specific contamination to work well when the reference benchmark uses a significantly different form than the original benchmark (e.g. multiple-choice instead of completion)?**
> The reviewer is correct to point out that an analysis would need to take this difference into account. For example, GSM8k (free-form) and MathQA (multiple-choice) are included in our experiments. We found that the Phi model family performed particularly badly on the MathQA dataset. This is likely due to the focus of the authors on textbook datasets during training. We believe the chain-of-thought capabilities of these models greatly increased because of this, but their ability to perform single-step mathematical computations, as required for MathQA, does not. This explains the very large discrepancy in performance between the two benchmarks. Therefore, conclusions here might not be indicative of information-flow-based contamination but do bring up a flaw in the model, i.e., its inability to perform well on mathematical operations without chain-of-thought reasoning.
>
> **What further research are the authors excited to see in this area? Are the authors planning on discussing this within the paper?**
> There are several exciting areas in data contamination that would benefit greatly from future research:
> - **Improving Information-Flow Based Contamination Detection**: Our method currently outperforms this traditional approach, but advancements here could provide more detailed contamination insights. For example, detecting specific contaminated samples could have applications in identifying the use of copyrighted materials.
> - **Extending ConStat to Multimodal Models**: Applying our approach to multimodal models, such as those involving images, would be valuable. Synthetic sample generation for complex data types like tabular images presents unique challenges, but overcoming these would increase the applicability of contamination detection.
> Unfortunately, due to space constraints, we did not discuss these topics in the NeurIPS submission. We will therefore not include it.
>
> We hope our answers address all of the reviewer’s concerns, remain happy to answer any further questions, and look forward to the reviewer’s response.

---

> > ### Comment · Reviewer_tnxc · 2024-08-12
> >
> > **How did you ensure that the generated dataset had a sufficiently high quality? Is the deduplication process using 1-grams sufficient?**
> > I accept the procedure presented by the authors is sufficiently rigorous. I would appreciate seeing this full procedure in the paper, as it could be useful for others who wish to replicate your work.
> >
> > **Why did the authors not select a reference benchmark for MMLU to measure benchmark-specific contamination?**
> > I accept your point.
> >
> > **Do you expect your approach for measuring benchmark-specific contamination to work well when the reference benchmark uses a significantly different form than the original benchmark (e.g. multiple-choice instead of completion)?**
> > I accept your point.
> >
> > **What further research are the authors excited to see in this area? Are the authors planning on discussing this within the paper?**
> > Interesting to hear about these ideas!

---

> > > ### Author Response · Authors · 2024-08-12
> > >
> > > Thank you for the positive reply! We will include the full description of our dataset generation process in the experimental details of the paper. We further note that we have included the full code-base in the supplementary material and plan to release it upon publication to ensure reproducibility.

---

### Official Review · Reviewer_E2e6 · 2024-07-13

**Soundness:** 3
**Presentation:** 3
**Contribution:** 2
**Rating:** 6
**Confidence:** 3

**Summary:**

This paper targets the problem of contamination detection of LLMs by proposing ConStat, a performance-based statistical approach.
- Instead of detecting the inclusion of test samples as contamination, the authors define interpretation as "abnormal" performance on benchmarks.
- ConStat builds on this definition, and leverages reference models and synthetic benchmarks for statistical testing at syntax-, sample-, and benchmark-specific levels.
- ConStat is empirically verified as effective. The authors also provide comprehensive analyses.

**Strengths:**

- The definition of contamination from a performance-based perspective is novel and interesting to my knowledge.
- Based on the definition, the proposed ConStat is intuitive and looks effective based on experiments. The experiments and analyses are comprehensive and solid.
- Contamination detection is timely and important in the current community.
- The paper is well-written and a joy to read.

**Weaknesses:**

- As the paper discussed in Section 6, it can only estimate the contamination relative to reference models.
- It would be a little costly to construct a synthetic dataset using close-sourced LLMs each time we want to detect contamination of a specific benchmark.
- This paper lacks some related works that need to be discussed. I just list a few below.

[1] Yang, Shuo, et al. "Rethinking benchmark and contamination for language models with rephrased samples." arXiv preprint arXiv:2311.04850 (2023).

[2] Jiang, Minhao, et al. "Investigating data contamination for pre-training language models." arXiv preprint arXiv:2401.06059 (2024).

**Questions:**

- According to lines 1118-1120, it seems that the authors only used 1000 samples for fine-tuning. How is this number determined and will the number of chosen samples affect the performance? if so, how?
- The above question applies to the number of synthetic data samples.

**Limitations:**

The authors offered a limitation paragraph in Section 6.

---

> ### Author Rebuttal · Authors · 2024-08-06
>
> We thank the reviewer for their detailed review and insightful questions. We are happy to hear that they find our paper a joy to read, our definition novel and interesting, and our experiments comprehensive and solid. Below, we address their questions.
>
> **Is measuring contamination relative to a set of reference models sufficient for contamination detection?**
> Please see our detailed answer in the main response.
>
> **Do we need to construct a synthetic dataset each time we want to measure contamination? If so, is this expensive?**
> No, we do not need to generate a new synthetic dataset each time we measure contamination. As long as our synthetic datasets remain private, they can be reused for future models. Of course, we will share these datasets with other interested parties if requested.
>
> A contamination analysis on other benchmarks would require us to create a new synthetic dataset. However, the cost of generating these datasets is relatively low compared to the total inference cost of model evaluation. Specifically, generating the synthetic dataset was only around five times as expensive as a single model evaluation. Since we evaluate 50 models on all benchmarks, the total cost of generating the synthetic dataset is much lower than the total inference cost of model evaluation. Finally, specifically for rephrasing, this task can be performed by cheaper open-weight models, further reducing costs.
>
> **Why have you not included citations to some recent works like Yang et al. and Jiang et al.?**
> Please see our detailed answer in the main response.
>
> **How does the number of samples in the synthetic dataset affect ConStat’s performance?**
> We choose the number of synthetic samples based on the trade-off between tight confidence bounds and computational budget. Computational complexity increases linearly with the number of samples, while the size of confidence intervals decreases. We found that 1000 samples provide tight confidence bounds and allow us to evaluate over 50 models within our budget.
>
> **Similarly, how does the number of samples used for fine-tuning affect results?**
> We again chose the number of samples used for fine-tuning in our ablation study and for comparison to other methods based on the trade-off between tight confidence bounds and computational complexity. While using more samples (a larger portion of the dataset) would yield tighter confidence bounds, it would also increase the cost of finetuning.
>
> We hope our answers address all of the reviewer’s concerns, remain happy to answer any further questions, and look forward to the reviewer’s response.

---

### Official Review · Reviewer_xz9Q · 2024-07-13

**Soundness:** 3
**Presentation:** 2
**Contribution:** 3
**Rating:** 5
**Confidence:** 4

**Summary:**

The paper introduces a new performance-based definition of data contamination, shifting the focus from the cause of contamination to its effect on performance. The paper also presents ConStat, a statistical method that detects and quantifies contamination by comparing performance on primary and reference benchmarks using a set of reference models. The effectiveness of ConStat is demonstrated through extensive evaluations on diverse model architectures, benchmarks, and contamination scenarios, revealing significant contamination in several popular models.

**Strengths:**

1. The motivation for proposing new definition of contamination is clear and easy to understand.
2. The experiments are extensive to demonstrate the effectiveness of the proposed method. It shows that ConStat can outperform the previous methods on contamination detection and is effective for contamination quantification.
3. The paper extends the analysis to the current model families to show that these models are contaminated in different degrees for current benchmarks. And the observation basically matches other papers' findings, suggesting that these methods can be potentially used for contamination examination of LLMs in the future applications.

**Weaknesses:**

1. The assumptions in the paper are too strong for me. I don't think the difference of performance in different benchmarks means that there would be contamination. The abnormal high performance in one dataset but normal performance in other benchmarks only imply higher possibility of contamination.
2. On line 152, the authors mentioned "additionally include an inherently uncontaminated random-guessing model". I wonder if more details could be provided about this.
3. I think the descriptions and writings in the methodology section should be re-written to provide more details and motivations for how the method is developed. The current version is somewhat hard to follow and lacks explanations.
4. Missing citations to many recent works. E.g., https://arxiv.org/abs/2311.04850 (Yang et al.), https://arxiv.org/abs/2401.06059 (Jiang et al.), etc.

**Questions:**

See the weaknesses above.

**Limitations:**

Yes, the authors included a limitations sections.

---

> ### Author Rebuttal · Authors · 2024-08-06
>
> We thank the reviewer for their detailed review and insightful questions. We are pleased to hear that they appreciate the extensive experiments and find the new definition clear and easy to understand. Below, we address their questions.
>
> **Does an abnormally high performance on your reference benchmark always imply information-flow based contamination on the actual benchmark?**
> Please see our detailed answer in the main response.
>
> **What is the random-guessing model that you add to the set of reference models?**
> The random-guessing model randomly guesses the answer to a query. It serves as an equivalent to including a very poor model in the reference set (e.g., gpt2-small). For open-form questions, this model's answers are always incorrect, resulting in 0% accuracy. For multiple-choice questions, it has a 1/k probability of being correct, where k is the number of options. Adding this model regularizes the hardness correction function's fit, ensuring it remains valid outside the score range of the other reference models. Figure 3f in Appendix B illustrates the impact of excluding the random-guessing model: without it, the hardness correction function overfits the reference models and fails to detect contamination accurately for weaker models. Including the random-guessing model prevents this issue.
>
> **Could you provide more details on the motivation for how the method is developed?**
> Certainly! In the methodology section, our goal is to develop a method that compares model performance across two related benchmarks and detect significant performance differences. Ideally, we would want to compute the performance of the model on both benchmarks and directly compare these results. We would like to conclude it is contaminated if the model scores high on one benchmark, but poorly on another. However, this straightforward approach is problematic because benchmarks often vary in difficulty, making direct performance comparisons unreliable. To address this, we introduce the hardness correction function, which maps performance from one benchmark to the corresponding expected performance on the other. This adjustment allows for performance comparisons, corrected for benchmark difficulty. The other parts of the section describe how to transform this method into a statistical test to calculate p-values for contamination.
> We note that all other reviewers found our paper well-written, with some even describing it as a joy to read. Thus, we believe our paper's structure and explanations are clear and precise. However, if the reviewer has further questions, we are more than happy to provide additional clarification.
>
> **Why have you not included citations to some recent works like Yang et al. and Jiang et al.?**
> Please see our detailed answer in the main response.
>
>
>
> We hope our answers address all of the reviewer’s concerns, remain happy to answer any further questions, and look forward to the reviewer’s response.

---

### Author Rebuttal · Authors · 2024-08-06

$\newcommand{R}{\textcolor{green}{E2e6}}$ $\newcommand{S}{\textcolor{blue}{tnxc}}$ $\newcommand{T}{\textcolor{purple}{hKkW}}$ $\newcommand{X}{\textcolor{red}{UkYs}}$ $\newcommand{Y}{\textcolor{brown}{xz9Q}}$

We thank the reviewers for their detailed reviews and insightful questions. We are pleased to hear that they found our paper well-written ($\R,\S,\T,\X$), our new definition of contamination novel and interesting ($\Y,\R,\X$), and our experiments comprehensive and solid ($\Y,\R,\S,\T$). We identified several questions shared among reviewers that we address here. Reviewer-specific questions are answered in the respective responses.


**1. Does an abnormally high performance on your reference benchmark always imply information-flow based contamination on the actual benchmark? ($\Y,\T$)**
Yes, we argue that in the case of syntax- and sample-specific contamination, an abnormally high performance always implies information-flow based contamination.

To demonstrate this, let us first assume that our synthetic benchmarks are drawn from the same distribution as the original benchmark. A non-contaminated model should achieve a similar performance on both benchmarks. Indeed, if the model obtains a much higher score on one benchmark, we could distinguish between samples from the synthetic and original benchmark with non-random probability, contradicting the assumption that samples are drawn from the same distribution. Therefore, a model that shows abnormally high performance on one of the benchmarks has to be contaminated with that benchmark.

Our synthetic datasets are designed to closely approximate the original benchmark distribution. Figure 2 shows that this approximation is highly accurate, as we can predict the exact performance difference between the original benchmark and another benchmark known to be drawn from the same distribution. Hence, syntax- and sample-specific contamination imply information-flow-based contamination.

In contrast, for benchmark-specific contamination, the reviewers are correct to point out that differences in performance do not always indicate information-flow based contamination. We acknowledge this in our paper and draw more cautious conclusions regarding this type of contamination. However, measuring benchmark-specific contamination remains important. Common practice often involves evaluating performance on only one or two benchmarks for a specific task, such as mathematics or coding. If a model excels on these benchmarks but performs poorly on others measuring the same task, we argue that this indicates problematic behavior. While this might not be information-flow-based contamination, it may result from a focus on a subset of the task during training because of a drive to perform well on a specific benchmark. Benchmark-specific contamination can reveal this issue and indicate how representative reported scores are with regard to overall task performance.


**2. Why have you not included citations to some recent works like Yang et al. and Jiang et al.? ($\Y,\R$)**
The papers mentioned are part of a large subfield in contamination detection for LLMs that assumes access to training data. This subfield analyzes contamination and develops efficient algorithms to detect benchmark samples in the extensive training datasets of current LLMs. However, these works strongly assume access to training data. This makes them only relevant for model providers since training data is rarely shared, even for open-weight models. In contrast, ConStat does not make this assumption. Furthermore, these works rely on additional assumptions about the training data to measure the influence of the contamination on performance. For instance, they cannot determine the contamination's impact on performance if all benchmark samples are included in the training data.

We were aware of this research, including both works, but did not discuss it due to its strong assumptions and inability to quantify contamination. Since two reviewers highlighted these works, we will update the paper to discuss them in our Related Work section, highlighting that they require access to the training data while ConStat does not.


**3. Is measuring contamination relative to a set of reference models sufficient for contamination detection? ($\R,\T$)**
Yes, as the primary goal of benchmarking is to compare the performance of different approaches/models accurately. Using these results for model selection and to track progress in the field only requires a relative performance measurement. For instance, a 5% performance increase across all models due to contamination will leave the model rankings untouched. Accurate absolute performance estimates primarily enable comparisons with a golden standard, such as human performance. While valuable, this is not the main goal of benchmarks, as can be seen by the fact that human performances are mostly ignored and are sometimes not even known. Thus, we argue that relative contamination measurement is almost as valuable as absolute measurement and worth pursuing.

---

### Decision · Program_Chairs · 2024-09-25

**Decision:**

Accept (poster)

**Comment:**

The paper proposes ConStat, a statistical method for detecting and quantifying contamination in large language models by comparing performance between a primary benchmark and a reference benchmark relative to a set of reference models.

The paper contributes: a new performance-based definition of contamination as "artificially inflated and non-generalizing benchmark performance" rather than just the inclusion of benchmark samples in training data; a novel method that can detect contamination in both open and closed-source models without access to training data. The paper also performs extensive evaluation demonstrating ConStat's effectiveness across diverse model architectures, benchmarks, and contamination scenarios. The paper finds high levels of contamination in several popular models like Mistral, Llama, Yi, and top Open LLM Leaderboard models.

**Strengths**: Novel performance-based approach to contamination detection; comprehensive experiments demonstrating effectiveness, especially for syntax and sample-specific contamination; generalization to both open and closed-source models.

**Weaknesses**: reliance on synthetic reference datasets, which may introduce errors; limited analysis of benchmark-specific contamination; arguably, the method detects performance differences rather than direct contamination, which could lead to false positives in some cases; the method only provides results that are relative to reference models, not absolute measures of contamination.

Overall, this is a solid paper with novel ideas and good empirical results.